# Galactosyl- and glucosylsphingosine induce lysosomal membrane permeabilization and cell death in cancer cells

Kamilla Stahl-Meyer[1,2], Mesut Bilgin[1], Lya K. K. Holland[1], Jonathan Stahl-Meyer[1], Thomas Kirkegaard[2], Nikolaj Havnsøe Torp Petersen[2], Kenji Maeda[1]*, Marja Jäättelä[1,3]*

1 Cell Death and Metabolism, Center for Autophagy, Recycling and Disease, Danish Cancer Society Research Center, Copenhagen, Denmark, 2 Orphazyme A/S, Copenhagen, Denmark, 3 Department of Cellular and molecular Medicine, Faculty of Health Sciences, University of Copenhagen, Copenhagen, Denmark

* mj@cancer.dk (MJ); kenjim@cancer.dk (KM)

**Data Availability Statement:** All relevant data are within the manuscript and its Supporting Information files.

## Abstract

Isomeric lysosphingolipids, galactosylsphingosine (GalSph) and glucosylsphingosine (GlcSph), are present in only minute levels in healthy cells. Due to defects in their lysosomal hydrolysis, they accumulate at high levels and cause cytotoxicity in patients with Krabbe and Gaucher diseases, respectively. Here, we show that GalSph and GlcSph induce lysosomal membrane permeabilization, a hallmark of lysosome-dependent cell death, in human breast cancer cells (MCF7) and primary fibroblasts. Supporting lysosomal leakage as a causative event in lysosphingolipid-induced cytotoxicity, treatment of MCF7 cells with lysosome-stabilizing cholesterol prevented GalSph- and GlcSph-induced cell death almost completely. In line with this, fibroblasts from a patient with Niemann-Pick type C disease, which is caused by defective lysosomal cholesterol efflux, were significantly less sensitive to lysosphingolipid-induced lysosomal leakage and cell death. Prompted by the data showing that MCF7 cells with acquired resistance to lysosome-destabilizing cationic amphiphilic drugs (CADs) were partially resistant to the cell death induced by GalSph and GlcSph, we compared these cell death pathways with each other. Like CADs, GalSph and GlcSph activated the cyclic AMP (cAMP) signalling pathway, and cAMP-inducing forskolin sensitized cells to cell death induced by low concentrations of lysosphingolipids. Contrary to CADs, lysosphingolipid-induced cell death was independent of lysosomal $Ca^{2+}$ efflux through P2X purinerigic receptor 4. These data reveal GalSph and GlcSph as lysosome-destabilizing lipids, whose putative use in cancer therapy should be further investigated. Furthermore, the data supports the development of lysosome stabilizing drugs for the treatment of Krabbe and Gaucher diseases and possibly other sphingolipidoses.

## Introduction

Glucosylsphingosine (GlcSph) and galactosylsphingosine (GalSph, also known as psychosine) are isomeric cationic lysosphingolipids composed of a sphingosine backbone and a hexosyl

**Funding:** This work was supported by grants from the Danish Cancer Society (R269-A15695), Danish National Research Foundation (DNRF125) and NovoNordisk Foundation (NNF19OC0054296) to M.J. and by the Independent Research Fund Denmark (6108–00542B) and Novo Nordisk Foundation (NNF17OC0029432) to K.M., and and by Innovation Fund Denmark (7038-00062B) to K. SM., N.H.T.P., and T.K.

**Competing interests:** The authors have declared that no competing interests exist.

headgroup [1, 2]. They exist at very low levels in healthy human cells and their normal functions are not known [3]. GlcSph and GalSph accumulate in the brain and other tissues of patients suffering from Gaucher and Krabbe diseases, respectively, and are thought as the primary causes of these pathologies [4]. Gaucher and Krabbe diseases are lysosomal storage diseases (LSDs), which include over 70 rare inherited monogenic diseases caused by mutations in genes encoding proteins essential for lysosomal function or transport [5, 6]. Gaucher disease is caused by biallelic mutations in the *GBA1* gene that encodes a lysosomal hydrolase, β-glucocerebrosidase (also known as glucosylceramidase, GBA1) [7]. GBA1 hydrolyses glucosylceramide (also known as glucocerebroside, GlcCer) to glucose and ceramide [7]. In addition to the primary substrate GlcCer, GlcSph accumulates in Gaucher disease likely owing to acid ceramidase-mediated hydrolysis of the N-bound fatty acid of GlcCer [2]. Krabbe disease is caused by mutations in the *GALC* gene encoding galactosylceramidase (also known as galactocerebrosidase, GALC) that degrades galactosylceramide (GalCer) to galactose and ceramide [1, 8]. The level of GalSph increases in Krabbe disease cells while that of GalCer is unaffected or even decreased [1, 9, 10]. The specific accumulation of GalSph in the brains of Krabbe disease patients has given a rise to the "Psychosine hypothesis" suggesting that the lysosphingolipid GalSph is the primary cause of the Krabbe disease pathology and that also the pathology of other sphingolipidoses, including Gaucher disease, is caused by the accumulation of lysosphingolipids rather than the primary substrates of the defective enzymes [1, 4]. However, the molecular mechanisms underlying the cytotoxicity of lysosphingolipids are yet unclear.

The lysosome-dependent cell death pathway contributes to several pathophysiological conditions such as neurodegeneration and cellular aging [11–13]. It is initiated by lysosomal membrane permeabilization, which leads to the leakage of lysosomal contents, including cathepsin proteases, into the cytosol [14–16]. Lysosomal leakage can be induced by several stimuli, including classical apoptosis inducers such as tumor protein p53 and tumor necrosis factor, which can permeabilize lysosomes either to initiate or to amplify their respective cell death pathways [17, 18]. Additionally, lysosomotropic agents such as amines with hydrophobic side chains (e.g. imidazole) [19], cationic amphiphilic drugs (CADs, e.g. siramesine and ebastine) [20–23], and sphingosine [24] are potent inducers of lysosomal leakage. Due to cancer-associated changes in lysosomal composition and decrease in lysosomal membrane stability [25, 26], several commonly used CADs and other lysosome-destabilizing agents are emerging as potential anti-cancer drugs [27]. Accordingly, the interest in the molecular mechanism underlying lysosomal membrane permeabilization is increasing, and recent data has revealed that its CAD-induced induction in cancer cells depends on a rapid, lysosomal $Ca^{2+}$ release through the P2X purinoreceptor 4 (P2RX4), subsequent synthesis of cyclic adenosine monophosphate (cAMP) by the $Ca^{2+}$-dependent adenylyl cyclase 1 (ADCY1) and the effective inhibition of lysosomal acid sphingomyelinase [20, 28]. The consequences of lysosomal leakage depend on its extent. Whereas extensive lysosomal damage leads to uncontrolled necrosis and moderate one initiates either caspase-dependent apoptosis or caspase–independent apoptosis-like cell death [18, 24, 29], spatially and temporally tightly controlled lysosomal leakage serves important, non-lethal functions in mitotic chromosome segregation, cell motility and inflammation [30, 31].

Prompted by a recent study by Folts and colleagues showing that cell death induced by GalSph and GlcSph is preceded by an increased lysosomal pH [32], and the fact that these lysosphingolipids share their amphiphilic nature and amine group with CADs [1, 2, 20], we asked whether GalSph and GlcSph induced lysosome-dependent cell death in a manner similar to that of CADs. Our data confirm that the lysosphingolipids induce cell death in breast cancer cells, which follows lysosomal leakage and shares many characteristics with CAD-induced lysosome-dependent cell death.

## Materials and methods

### Reagents

Chemicals, reagents, siRNAs and antibodies used are listed in S1–S3 Tables. All drugs were dissolved in DMSO unless otherwise specified.

### Cell cultures

All cells were cultured at 37˚C in a humidified atmosphere of 5% $CO_2$, and regularly tested and found negative for mycoplasma using Venor®GeM Classic PCR kit.

### MCF7 cells

The MCF7 cell line used in this study is a TNF-sensitive subclone (MCF7-S1) of the MCF7 human ductal breast carcinoma cell line [33]. The CAD-resistant MCF7 cell line was established and validated previously [28]. In short, MCF7 cells were cultured with increasing concentration of siramesine for 6 months. After thawing, CAD-resistant MCF7 cell were treated with 6 μM siramesine for 48 hours followed by 72 hours in the absence of the drug prior to experiments. The MCF7 galectin-3-eGFP cells were established and validated previously [34]. In short, prior to single cell sorting, MCF7 cells were transfected with galectin-3-eGFP plasmid using Lipofectamine LTX with PLUS reagent and selected with 600 μg/mL G418. All MCF7 cells were cultured in Dulbecco's Modified Eagle's medium (DMEM) supplemented with 6% (*v/v*) heat-inactivated fetal calf serum (FCS) and 1X penicillin and streptomycin.

### Fibroblasts

Skin fibroblasts derived from a healthy donor (GM00498) and a Niemann-Pick type C patient (GM18453) were purchased from the Coriell Institute. All fibroblasts were cultured in DMEM supplemented with 12% FCS, 1X non-essential amino acids, and 1X penicillin and streptomycin.

### Treatments

All cells were seeded 24 hours prior to any treatments if not otherwise indicated. Forskolin was added to the cells after 6 hours of treatment with indicated concentrations of GalSph, GlcSph, siramesine, or ebastine. Z-VAD-FMK, necrostatin-1, and ferrostatin-1 were added to subconfluent cells 1 hour prior to addition of GalSph, GlcSph, or ebastine. Cholesterol dissolved in ethanol was diluted to 60 μM in the cell culture medium by shaking at 37˚C for 1 hour, and added to the cells 24 hours before the addition of GalSph, GlcSph, or ebastine, which diluted the cholesterol concentration to 30 μM.

### Cell death assays

**Propidium iodine (PI) and Hoechst-33342 staining.** Subconfluent cells were treated as indicated, and cell death was measured using Celigo® Imaging Cytometer (Nexcelom Bioscience) after 10 min treatment of the cells with 0.2 μg/mL PI and 2.5 μg/mL Hoechst-33342 at 37˚C in a humidified atmosphere of 5% $CO_2$. In short, the instrument set the exposure time and focus for PI-staining, Hoechst-33342-staining, and the transmitted light channel. Then, images were taken of each selected well. For data analysis, masks of PI-positive cells and Hoechst-33342-positive cells were created based on the intensity and size of the stained cells. False-positive cells were eliminated by adjustments of these parameters, which were double-checked with the use of the transmitted light images. The count of the masks of Hoechst-

33342-positive cells provides the numbers of total cells, and the count of the masks of PI-positive cells the numbers of dead cells.

## Transfections

MCF7 cells were reverse transfected using 20 nM of siRNA (S3 Table) mixed with Lipofecta-mine™ RNAiMAX and optiMEM following the manufacturer´s instruction. In short, siRNA, Lipofectamine™ RNAiMAX, and optiMEM were mixed for 20 min and added to the MCF7 cells in suspension. Then, MCF7 cells were seeded on 96-well plate (6,000 cells per well) and the indicated treatments were initiated 24 hours later. Cell death was measured 72 hours after transfection.

## Western blotting

Proteins were separated by gradient SDS-PAGE gels (4–15%) and transferred to nitrocellulose membranes. Proteins were detected using indicated primary antibodies and appropriate horse-radish peroxidase-conjugated secondary antibodies following standard protocols (S2 Table). The blots were developed using Clarity™ Western ECL (BioRad) and images of the membranes were acquired using the luminescent image analyser LAS-4000 mini (Fujifilm, GE healthcare).

## Immunochemistry

The galectin puncta assay was used to visualize lysosomal membrane permeabilization as described previously [34, 35]. In short, cells grown in Greiner #655090 96-well plates were fixed in 4% paraformaldehyde in DPBS for 10 min and washed with 0.2% (*v/v*) triton X-100 in DPBS for 15 min. Then, cells were treated with 50 mM ammonium chloride for 10 min to quench background staining, and cells were permeabilized with ice cold methanol for 10 min at -20˚C. Next, cells were blocked for 30 min in Buffer 1 (1% (*w/v*) BSA, 0.3% (*v/v*) Triton X-100, 1 mM sodium azide, 5% (*v/v*) goat serum in DPBS) before staining with primary antibodies (S2 Table) diluted in Buffer 1 followed by appropriate AlexaFluor™488- or Alexa-Fluor™568-coupled secondary antibodies for 45 min. Lastly, DNA was labelled with 5 mg/mL Hoechst-33342 and cells were washed and left in DPBS. Images were acquired at the ImageX-press Micro Confocal system and analyzed using the MetaXpress analysis software according to the manufactory's instructions. Live cell imaging was performed on a MCF7 cell line stably expressing eGFP tagged LGALS3 (MCF7 galectin-3-eGFP). Subconfluent cells were treated with 0.1 μM Nuclear Violet and 0.2 μM SiR-Tubulin 2 hours prior to measurement in order to stain the nucleus and cytoplasm, respectively. Images were acquired using ImageXpress in widefield mode using a 60X air objective. Image analysis was performed in the MetaXpress analysis software. Here, an image analysis pipeline was created in its custom module. Images were exposed to cell segmentation, where the nuclei and cytoplasm were masked based on fluorescent intensity and size of Nuclear Violet and SiR-Tubulin, respectively. Then, a cell was defined as being positive for both a nuclei and cytoplasm mask. Border objects were removed to analyse whole cells. Galectin puncta were counted by detecting "round objects" based on fluorescent intensity and pixel size inside a defined cell. A "top hat" module was added to detect true puncta and to exclude artefacts arising from fluorescent accumulation at the edge of the cells. The analysis output was the total cell number, cells positive for galectin puncta (defined as a cell with at least one galectin puncta), and the number of galectin puncta in each galectin puncta positive cell.

## Measurement of lysosomal pH

A total of 45,000 MCF7 cells were seeded in an 8-well slide (Thermo Fisher Scientific #155361) and cultured for 8 hours. The cells were then treated for 16 hours with 1.25 mg/mL fluorescein (FITC)- and tetramethylrhodamine (TMR)-coupled dextran (Thermo Fisher Scientific) to allow them to enter lysosomes via endocytosis. The cells were then washed and fresh medium was added for a 1-hour chase period before being left untreated (DMSO) or treated with 10 nM concanamycin A (Santa Cruz, Dallas, TX, USA), 53 μM GalSph or 40 μM GlcSph for 1 or 5 hours. The medium was aspirated and cells washed once and fresh imaging solution was added (Thermo Fisher Scientific). The images were acquired immediately after on a LSM800 confocal microscope by acquiring FITC and TMR simultaneously using a plan-Apochromat 63×/1.40 Oil DIC M27 objective and Zen 2010 software (Carl Zeiss). Data was analyzed in ImageJ (Java 1.8.0_172 [64-bit]) and Excel (16.0.5332.1000), and illustrations made in Prism Software (8.0.2) and Adobe illustrator.

## Sample preparation for lipidomics

Cells detached from cell culture dishes using TrypLE were washed three times with 1 mL ice-cold 155 mM ammonium bicarbonate by spinning them down at 500 g for 5 min at 4°C and re-suspended in 155 mM ammonium bicarbonate to 3,000 cells/μL. Lysosomes were purified using a method based on magnetic column chromatography of lysosomes loaded with iron dextran particles (FeDEX), as described previously [36]. In short, cells were cultured for 16 hours in a medium supplemented with in-house prepared iron dextran particles (FeDEX), and then chased in fresh medium without FeDEX for at least three hours. The following procedures were performed at 4°C. Harvested cells were lysed and the lysosomes were captured by loading the cell lysates on an MS column (Miltenyi Biotec, Germany) mounted on a magnetic separator. The lysosomes retained on the column were eluted using a plunger after the column was removed from the magnetic separator. The purified lysosomes were pelleted by centrifugation at 21,000 g for 30 min and re-suspended in 200 μL 155 mM ammonium bicarbonate.

## Lipid extraction and shotgun lipidomics analyses

Lipid extraction and shotgun lipidomics analyses were performed as described previously [37]. Lipids were extracted from a suspension of 200,000 cells or from purified lysosomes suspended in 200 μL 155 mM ammonium bicarbonate using a modified Bligh and Dyer protocol [38, 39]. The samples in Eppendorf tubes were spiked with appropriate amounts of internal lipid standards (S4 Table), and lipids were extracted by adding 1.0 mL chloroform/methanol (2:1, *v/v*) and thoroughly shaking the mixtures for 20 min at 4°C. The mixtures were centrifuged at 1,200 g for 2 min at 4°C, and the lower phase containing the extracted lipids were transferred to new Eppendorf tubes. The lipid extracts were dried in a vacuum centrifuge for approximately 1 hour and re-dissolved in chloroform/methanol (1:2, *v/v*). Then, samples were mixed with positive or negative ionization solvents (13.3 mM ammonium bicarbonate in 2-propanol or 0.2% (*v/v*) tri-ethyl-amine in chloroform/methanol (1:5, *v/v*) and analyzed in positive and negative ion modes on quadrupole-Orbitrap mass spectrometer Q Exactive (Thermos Fisher Scientific, Waltham, MA, USA) equipped with TriVersa NanoMate (Advion Biosciences, Ithaca, NY, USA) for automated and direct nanoelectrospray infusion [40, 41]. The acquired data were analyzed using the LipidXplorer software to identify lipids [42, 43]. Lipids were annotated with their sum composition and the classification of lipid classes was according to LIPID-MAPS [44] (S5 Table). The identified lipids were quantified using an in-house developed script LipidQ (https://github.com/ELELAB/lipidQ) [41].

## Statistics

Data is presented as mean ± standard deviation (SD) based on at least three independent experiments. The statistical analyses have been conducted using the Prism 8 software to test the null hypothesis, which is a hypothesis that proposes that there is no significant difference between two variables in the hypothesis. The significance level (alpha) was set to 0.05. Two-way ANOVA has been used in order to determine how the measurement (e.g. cell death) is affected by two factors. When the measurement has been compared between a treatment with different concentrations in two different cell lines, the two-way ANOVA followed by Sidak's multiple comparisons test has been conducted to test the null hypothesis. When the measurement has been compared between the controls and different treatments in one cell line, the two-way ANOVA followed by Dunnett's multiple comparisons test has been conducted to test the null hypothesis. When one measurement is compared in one cell line between one control and different measurements, the one-way ANOVA followed by Dunnett's multiple comparisons test has been conducted to test the null hypothesis. In order to analyse the development of puncta with many time points (Fig 3), the null hypothesis of changes in the number of puncta over time are the same for the different treatments was tested. First, the area under the curve (AUC) was quantified, and then applied to an unpaired, since the measurements did not come from the same cell, two-tailed t-test with Welsh's correction when the variances between the treatments were significantly different determined by the F-test. Lipidomics statistics were conducted using unpaired t-test with Welch's correction for each lipid class. This was done in order to compare the mean of mol% between control and treatments in each lipid class, which are unmatched groups. Here, it is assumed that the values follow a normal distribution and with Welch's multiple comparison test, the analysis does not assume the values to have the same standard deviation in between the treatments. P values are shown as $P < 0.05 = ^*$, $P < 0.01 = ^{**}$, $P < 0.001 = ^{***}$ and $P < 0.0001 = ^{****}$.

## Results

### GalSph and GlcSph kill MCF7 cells independent of apoptosis, ferroptosis, and necroptosis

To study the mechanism of lysosphingolipid-induced cell death, we first defined the efficacy of GalSph and GlcSph to induce plasma membrane permeabilization in subconfluent MCF7 cells grown in complete growth medium and treated with the lipids for 48 hours. Akin to CADs, siramesine and ebastine, GalSph and GlcSph induced cell death in a dose-dependent manner with relatively steep dose response curves. The lethal concentration 50 (LC50) values of GalSph and GlcSph were 44 μM and 33 μM, respectively, *i.e.* significantly higher than those of siramesine (7 μM) and ebastine (15 μM) defined in parallel (Fig 1A).

To define the role of apoptotic caspases in lysosphingolipid-induced cell death, we tested the ability of Z-VAD-FMK, a well-characterized pan-caspase inhibitor [15], to rescue the cells. Treatment of MCF7 cells with 10 or 20 μM Z-VAD-FMK had no effect on either GalSph- or GlcSph-induced cytotoxicity, while both concentrations effectively inhibited cell death induced by an apoptosis-inducing drug staurosporine (Fig 1B and 1C) [45]. In line with this, treatment of MCF7 cells with lysosphingolipids for up to 20 hours failed to activate effector caspases as defined by their inability to induce the cleavage of caspase-7, the main effector caspase in MCF7 cells, or its substrate poly(ADP-ribose)polymerase (PARP) (Fig 1D) [15, 46, 47]. As expected, caspase-7 and PARP were effectively cleaved in staurosporine-treated MCF7 cells (Fig 1D). In addition to the inhibition of apoptosis by Z-VAD-FMK, lysosphingolipid-induced cell death was insensitive to necrostatin-1, an inhibitor of necroptosis-inducing receptor-

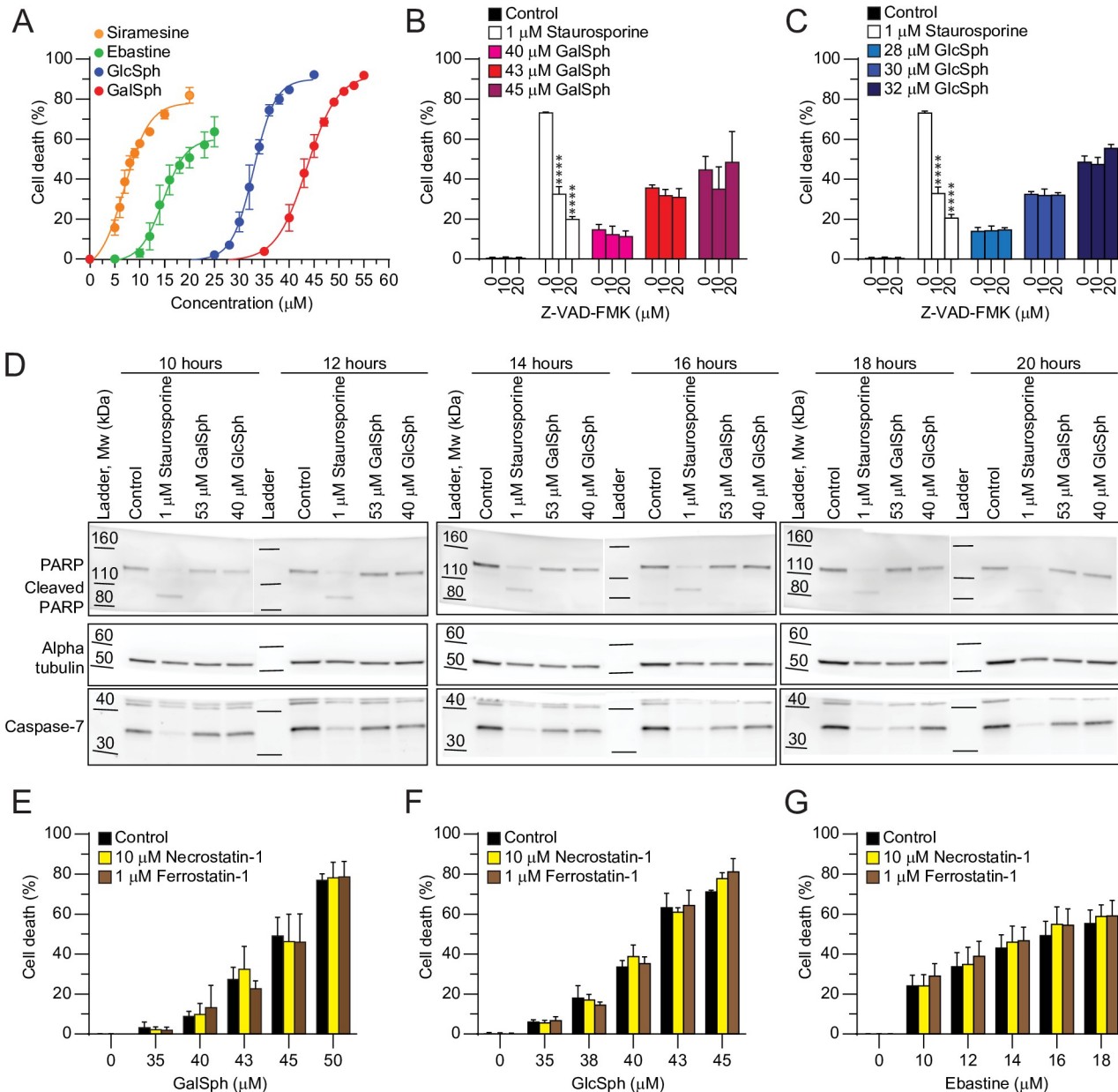

**Fig 1. GalSph- and GlcSph-induced death of MCF7 cells is independent of apoptosis, necroptosis and ferroptosis.** (A) Death of MCF7 cells treated for 48 hours with indicated concentrations of GalSph, GlcSph, ebastine or siramesine was determined by propidium iodine and Hoechst-33342 staining employing Celigo Imaging Cytometer. Connecting lines represent non-linear fit with $R^2 > 0.99$ for each treatment. (B, C) Death of MCF7 cells treated for 48 hours with indicated concentrations of GalSph (B) or GlcSph (C) after 1 hour pre-treatment with indicated concentrations of Z-VAD-FMK was determined as in (A). Staurosporine served as a positive control for apoptotic cell death. (D) Representative Western blots of PARP, caspase-7, and alpha tubulin (loading control) in lysates of MCF7 cells treated for indicated times with GalSph or GlcSph at concentrations corresponding to their LC90 values. Staurosporine served as a positive control for apoptotic cell death. (E-G) Death of MCF7 cells treated for 48 hours with indicated concentrations of GalSph (D), GlcSph (E), or ebastine (F) after 1 hour pre-treatment with indicated concentrations of necrostatin-1 or ferrostatin-1 was determined as in (A). Error bars, SD of three independent experiments. *, $P < 0.05$; **, $P < 0.01$; ***, $P < 0.001$; ****, $P < 0.0001$ as analyzed by two-way ANOVA followed by Dunnett's (B-C) or Sidak's (E-G) multiple comparisons tests.

interacting serine/threonine-protein kinase 1, and ferrostatin-1 that inhibits ferroptosis by trapping peroxyl radicals (Fig 1E–1G) [48, 49].

Taken together, these data suggest that GalSph and GlcSph activate a cell death pathway distinct from apoptosis, necroptosis, and ferroptosis in MCF7 cells.

## GalSph and GlcSph induce lysosomal leakage prior to plasma membrane permeabilization

We have previously established a subline of MCF7 cells that is partially resistant to lysosome-dependent cell death induced by a variety of CADs [28]. Interestingly, these cells showed a similar partially resistant phenotype against cell death induced by GalSph and GlcSph (Fig 2A–2D).

Akin to CADs, GalSph and GlcSph effectively inhibited the growth of MCF7 cells even at sublethal concentrations (Fig 2E–2H). Contrary to cell death, CAD-resistant cells were as sensitive to the growth inhibitory effects of CADs and lysosphingolipids as parental cells (Fig 2E–2H). These data suggest that lysosphingolipids may induce a cell death similar to CAD-induced lysosome-dependent cell death.

The lysosome-dependent cell death is characterized by the occurrence of lysosomal membrane permeabilization prior to the loss of plasma membrane integrity [15]. To investigate whether lysosphingolipids destabilize lysosomes, we took advantage of MCF7 cells expressing galectin-3-eGFP, which forms lysosomal galectin-3-eGFP puncta upon lysosomal membrane rupture [34]. Untreated MCF7 galectin-3-eGFP cells had a diffuse cytosolic distribution of galectin-3-eGFP throughout the 13-hour follow-up (Fig 3A).

Treatment of these cells with lysosphingolipids at concentrations that killed 60–80% of the cells in 48 hours (50–53 µM GalSph; 38–40 µM GlcSph) resulted in a significant increase in the percentage of cells with galectin-3 puncta already 8–9 hours after the treatment, whereas significant increase in cell death was observed first after 20 hours (Fig 3A–3D). Lysosphingolipid-induced appearance of galectin-3 puncta was, however, clearly slower, and weaker than that induced by 12 or 16 µM ebastine (Fig 3A–3D). Unlike CADs [50], the lysosphingolipids did not induce notable elevation in the lysosomal pH in MCF7 cells after one or five hours of treatments and thus prior to lysosomal leakage, when assessed after loading lysosomes with dextran coupled with pH-sensitive fluorescein and pH-insensitive tetramethylrhodamine (S1 Fig). Taken together, these data indicate that GalSph and GlcSph induce lysosomal membrane permeabilization prior to the loss of plasma membrane integrity, which suggests that they activate lysosome-dependent cell death.

## Exogenously supplied GalSph and GlcSph enter the lysosomes

Next, we investigated how much of the applied lysosphingolipids enters the cells and whether they accumulate in lysosomes. For this purpose, we used mass spectrometry-based shotgun lipidomics [41], which does not differentiate between the isomeric GalSph and GlcSph but detects both as hexosylsphingosines (HexSph). First, we treated MCF7 cells with 45 µM GalSph or 32 µM GlcSph and monitored HexSph levels in the cell culture medium and in cell lysates immediately after the treatment (1 min) and 1, 3, 6 and 24 hour later. Both lysosphingolipids were effectively taken up by the cells. The HexSph concentration in the cell culture medium decreased by over 40% after one hour of treatment and by over 60% after 24 hours (Fig 4A and 4B).

In parallel, the level of HexSph in the harvested cells increased rapidly during the first hour of treatments with GalSph or GlcSph and continued to increase for 24 hours (Fig 4A and 4B). The molar percentage (mol%) of HexSph of the total molar quantity of all monitored lipids (26 lipid classes, including the highly abundant cholesterol and phosphatidylcholine [51]) reached 5–10 mol% already after 1 hour and approximately 20% after 24 hours of treatment (Fig 4C).

To define how much of the exogenously supplied lysosphingolipids reached the lysosomes, we loaded the lysosomes of MCF7 cells with iron dextran (FeDex) particles prior to the indicated treatments, purified FeDex containing lysosomes using magnetic column

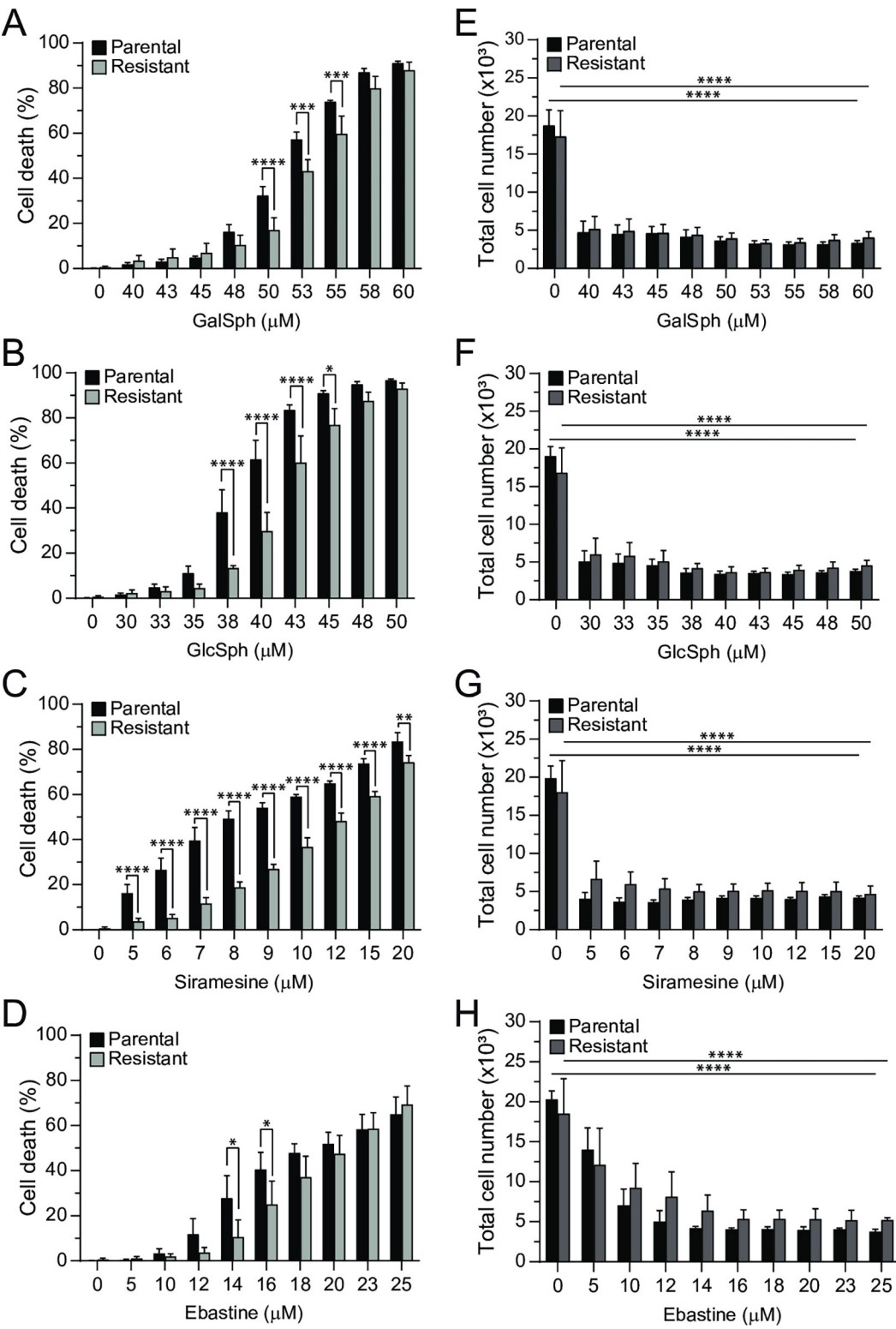

**Fig 2. CAD-resistant MCF7 cells are resistant against GalSph- and GlcSph-induced cell death but not to growth inhibition.** (A-D) Death of parental and CAD-resistant MCF7 cells treated for 48 hours with indicated concentrations of GalSph (A), GlcSph (B), siramesine (C, positive control), or ebastine (D, positive control) was determined as in in Fig 1A. (E-H) Total cell numbers of parental and CAD-resistant MCF7 cells treated with indicated concentrations of GalSph (E), GlcSph (F), siramesine (positive control) (G), or ebastine (positive control) (H) for 48 hours were determined by Hoechst-33342 staining employing Celigo Imaging Cytometer. Error bars, SD of three independent experiments. *, $P < 0.05$; **, $P < 0.01$; ***, $P < 0.001$; ****, $P < 0.0001$ as analyzed by two-way ANOVA followed by Sidak's multiple comparisons tests.

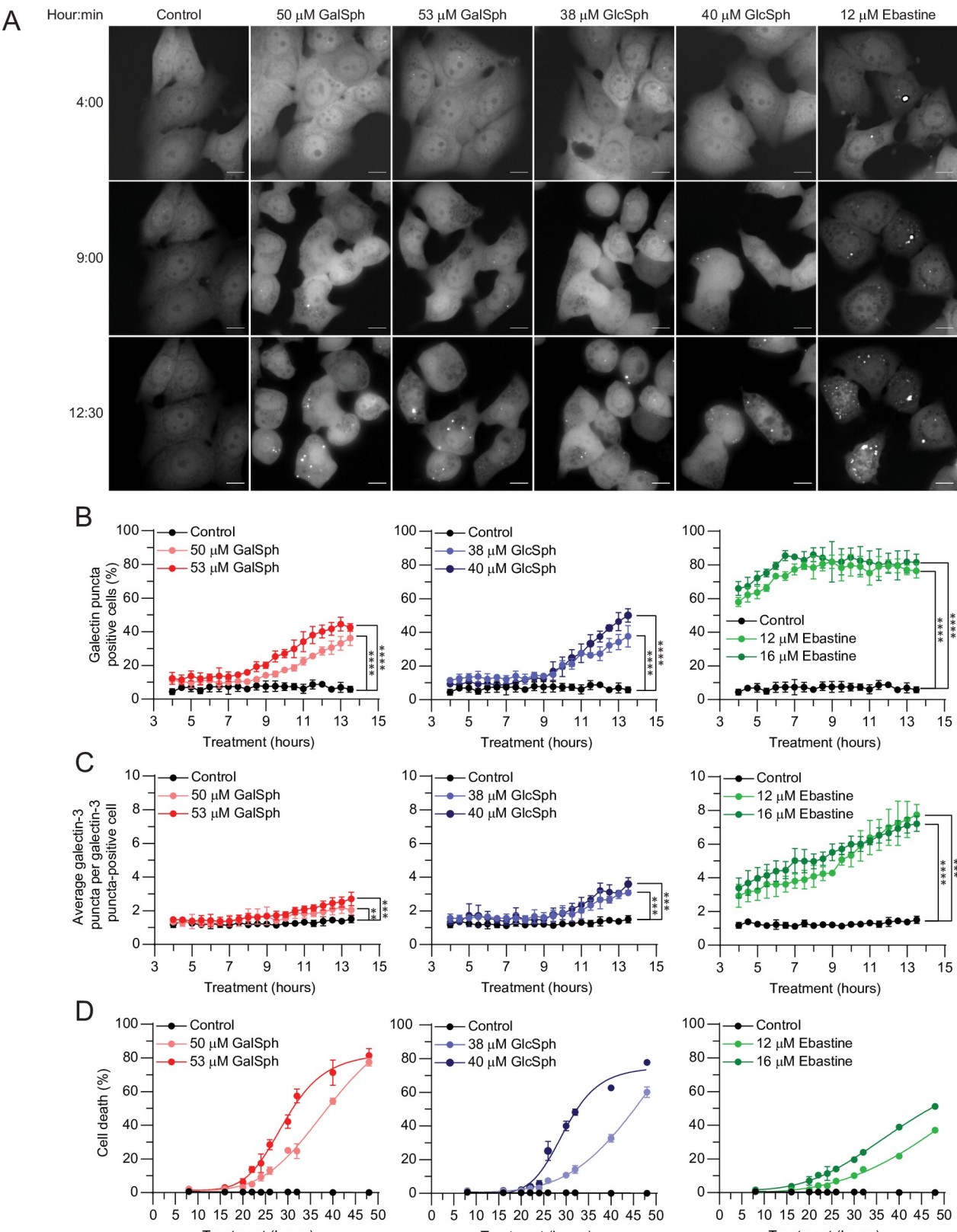

**Fig 3. GalSph and GlcSph induce lysosomal membrane permeabilization in MCF7 galectin-3-eGFP cells.** (A) Representative images of MCF7 galectin-3-eGFP cells treated with indicated concentration of GalSph, GlcSph, or ebastine for indicated times (hour:min) were acquired using the

ImageXpress (*FITC* channel). Scale bars, 10 μm. (B) Percentage of galectin-3 puncta positive MCF7 galectin-3-eGFP cells treated as indicated with GalSph, GlcSph, or ebastine and visualized as in (A). (C) Average numbers of galectin-3 puncta in puncta-positive MCF7 galectin-3-eGFP cells treated as indicated with GalSph, GlcSph, or ebastine and visualized as in (A). (D) Death of MCF7 cells treated with GalSph, GlcSph, or ebastine for indicated times (n = 3). Connecting lines present the non-linear fit calculated by GraphPad Prism. Error bars, SD of three independent experiments with >100 randomly chosen cells analyzed *per* sample. **, *P* < 0.01; ***, *P* < 0.001; ****, *P* < 0.0001 as analyzed by unpaired t-test of the area under the curve (AUC) values with Welsh's correction.

chromatography [51], and quantified total cellular and lysosomal HexSph by shotgun lipidomics. As expected [3], lysates and lysosomal fractions of untreated MCF7 cells had barely detectable amounts of HexSph (Fig 4D). The treatment of cells with 50 μM GalSph or 40 μM GlcSph for 3 hours resulted in similar relative quantities of HexSph (approximately 30 mol%) in entire cells and lysosomes suggesting that the exogenously added lipids reached the lysosomes but did not specifically accumulate in them at this early time point (Fig 4D). Confirming the quality of the lysosomal purification, lysosomal lipidome (excluding HexSph) contained 7–10 mol% of lysosome-specific phospholipid, bis(monoacylglycero)phosphate (BMP) (isobaric to phosphatidylglycerol (PG); referred here to as BMP/PG), in comparison to below 0.5 mol% BMP/PG in the lipidome of whole cell lysates (Fig 4E). Accordingly, the relative amount of mitochondria-specific cardiolipin (CL) in lysosomal and cellular lysates was < 0.1 mol% and approximately 5 mol%, respectively (Fig 4F). In line with a reported ability of GlcSph to interfere with phosphatidylcholine (PC) metabolism by inhibiting its synthesis

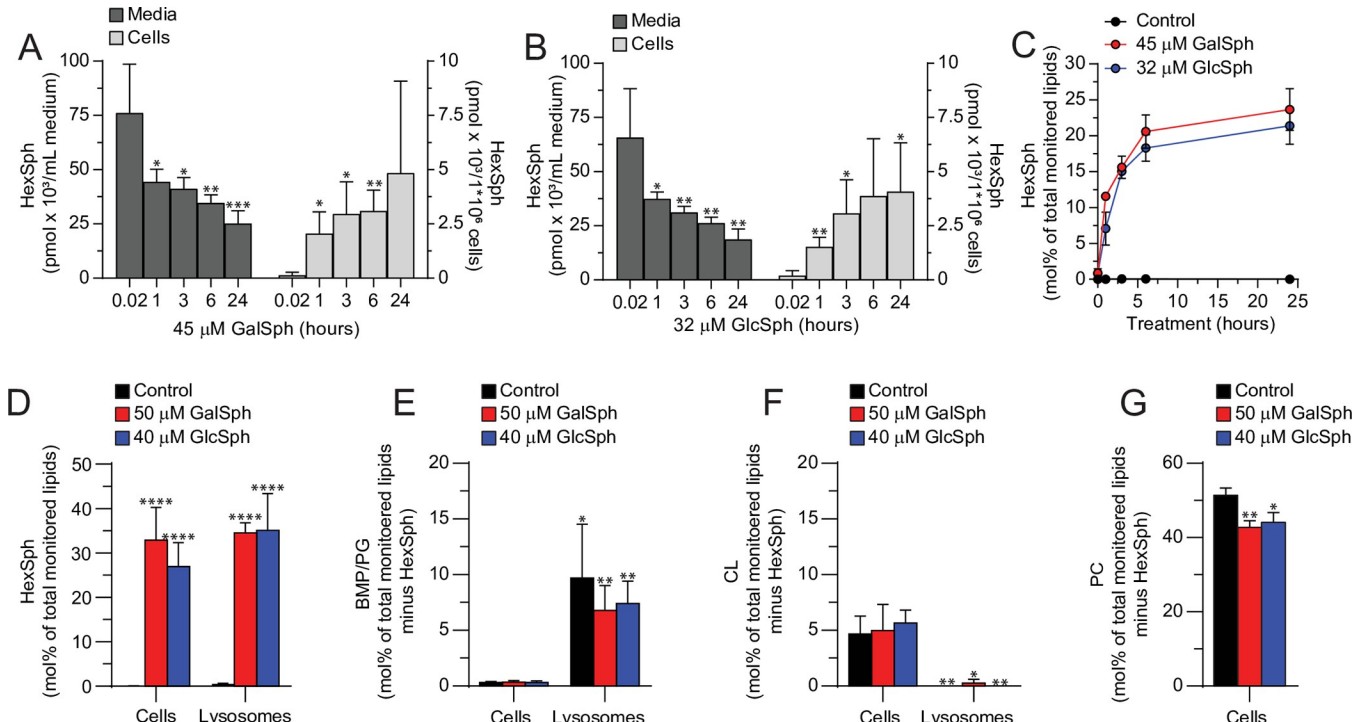

**Fig 4. Extracellular GalSph and GlcSph enter the lysosomes of MCF7 cells.** (A, B) Quantitites of HexSph (GalSph and GlcSph) in cell lysates (cells) and cell culture medium (media) of MCF7 cells treated with 45 μM GalSph (A) or 32 μM GlcSph (B) for indicated times were analyzed by shotgun lipidomics. (C) The molar percentages (mol%) of HexSph in MCF7 cells treated with 45 μM GalSph or 32 μM GlcSph for indicated times were analyzed as in (A). (D-G) The molar percentages of HexSph (D), BMP/PG (E), CL (F), and PC (G) lipid classes in total cell lysates (cells) and in purified lysosomes of MCF7 cells treated as indicated for 3 hours were analyzed as in (A). Error bars, SD of three independent experiments. *, *P* < 0.05; **, *P* < 0.01; ***, *P* < 0.001, ****; *P* < 0.0001 comparing the treated values to the first measurement at 0.02 hours (A, B) or to the control appropriate sample (D-G) and analyzed by unpaired t-test (A, B, E-G) or two-way ANOVA followed by Sidak's multiple comparisons test (D). BMP, bis(monoacylglycero)phosphate; PG, phosphatidylglycerol; CL, cardiolipin; PC, phosphatidylcholine.

by cytidylyltransferase and by activating its degradation by phospholipase D [52], treatment of MCF7 cells with GlcSph or GalSph significantly reduced cellular PC levels (Fig 4G).

The lipidomics data presented above indicate that lysosphingolipids added to the culture medium of MCF7 cells enter the cells, reach lysosomes in high quantities, and alter cellular lipid metabolism.

## GalSph and GlcSph activate cAMP signalling pathway required for CAD-induced lysosome-dependent cell death

CAD-induced lysosome-dependent cell death depends on the activation of the cyclic AMP (cAMP) signalling pathway [28]. Thus, we investigated the ability of GalSph and GlcSph to activate this pathway in MCF7 cells by analysing the phosphorylation status of serine-133 in the cAMP-responsive element-binding protein (CREB) [28]. Both lysosphingolipids induced a similar increase in phosphorylated CREB (P-CREB) as CADs, albeit with slower kinetics (Fig 5A).

Whereas CAD-induced P-CREB reached maximal levels already 4 hours after the treatment, the lysosphingolipid-induced increase was evident first after 6 hours, whereafter it remained at high levels throughout the 12-hour follow-up (Fig 5A). Supporting the role of cAMP in GalSph and GlcSph cytotoxicity, further activation of cAMP synthesis by forskolin, a potent activator of adenylate cyclase, sensitized MCF7 cells to cell death induced by low concentrations of lysosphingolipids in a similar manner as it sensitized cells to CADs (Fig 5B–5E). Forskolin failed, however, to increase cell death of MCF7 cells in combination with higher concentrations of the lysosphingolipids (Fig 5B and 5C). This biphasic effect of forskolin in cells treated with GalSph or GlcSph suggests that lysosphingolipids may induce cell death through different pathways depending on their concentration.

CAD-induced increase in intracellular cAMP levels and lysosome-dependent cell death depends on lysosomal $Ca^{2+}$ release through P2RX4 purinoreceptor [28]. The effective siRNA-mediated depletion of P2RX4 in MCF7 cells, which significantly reduced siramesine-induced cell death, had, however, no effect on the lysosphingolipid-induced cell death (Fig 5F–5I). Taken together, this indicates that even though lysosphingolipids and CADs induce a similar activation of the cAMP pathway, the early events in their respective cell death pathways are distinct.

## Cholesterol protects against lysosphingolipid-induced cell death

Cholesterol has a stabilizing effect on membranes, and cholesterol overload is an effective means to inhibit lysosomal leakage [30, 53]. To investigate whether lysosphingolipid-induced cell death was dependent on lysosomal membrane permeabilization, we treated MCF7 cells with 60 μM cholesterol for 24 hours prior to the addition of 40–50 μM GalSph or 25–32 μM GlcSph and throughout the 48-hour incubation with lysosphingolipids. The exogenously added cholesterol protected MCF7 cells almost completely against lysosphingolipid-induced cell death, even at highest lysosphingolipid concentrations that killed approximately 80% of the cells in the absence of cholesterol (Fig 6A and 6B).

As expected, cholesterol conferred also significant, albeit partial, protection against ebastine-induced cell death (Fig 6C). Furthermore, the exogenously added cholesterol showed a clear trend of reduced percentage of galectin puncta positive cells upon GlcSph treatment (Fig 6D and 6E). To investigate whether the strong cholesterol-mediated protection against lysosphingolipids was associated with reduced cellular uptake of the lipids, we measured HexSph levels in the harvested cells and in the cell culture medium. Cholesterol treatment did, however, not affect significantly the cellular uptake of either GalSph or GlcSph (Fig 6F and 6G). In

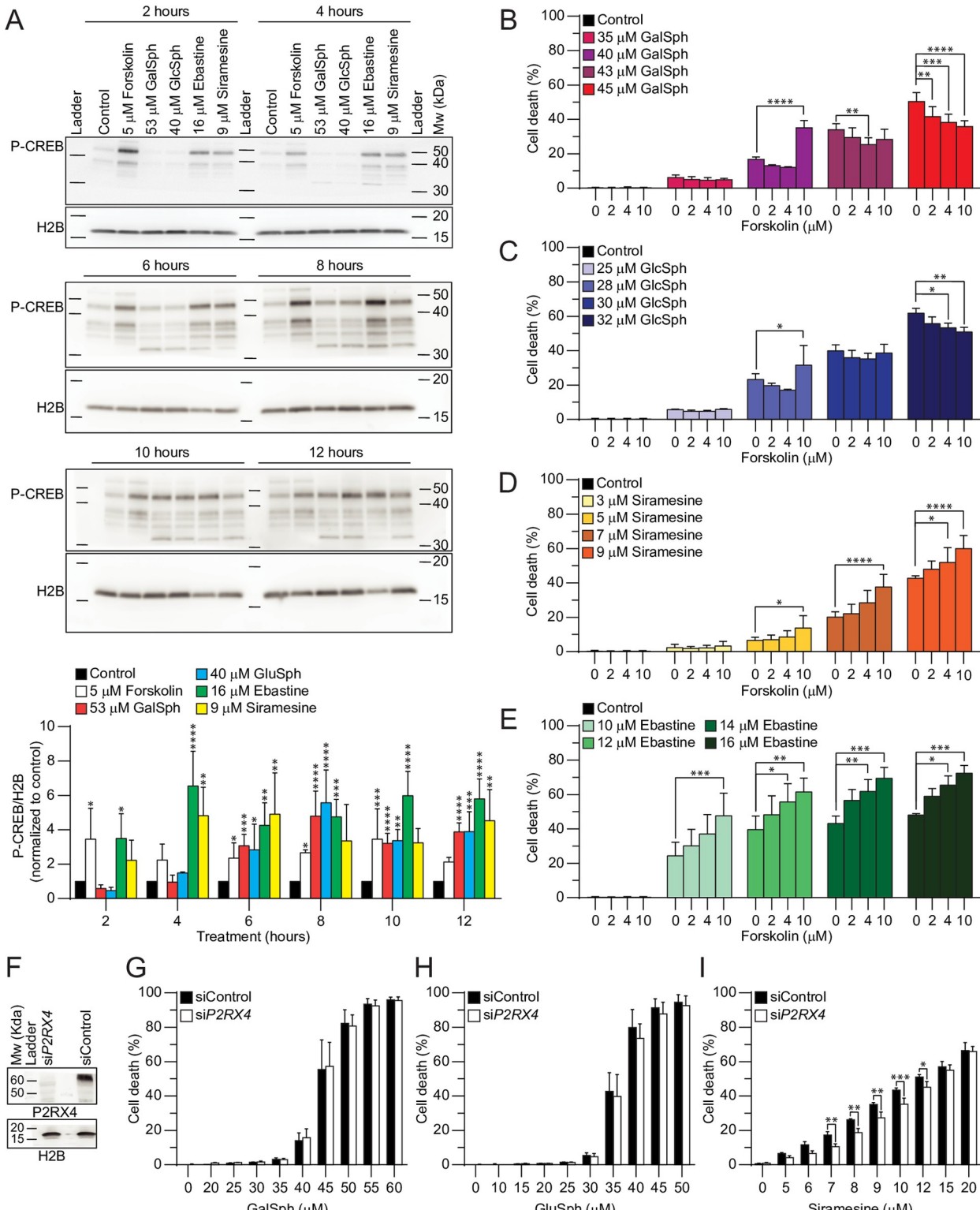

**Fig 5. GalSph and GlcSph increase cAMP levels in MCF7 cells.** (A) Representative Western blots of P-CREB and histone H2B (loading control) in lysates of MCF7 cells treated as indicated (*three top panels*), and densitometric quantification of P-CREB/H2B ratios normalized to untreated control samples (*bottom*). (B-E) Death of MCF7 cells treated with indicated concentrations of GalSph (B), GlcSph (C), siramesine (D, positive control), or ebastine (E, positive control) for 48 hours was determined as in Fig 1A. When indicated, 2–10 μM forskolin was added 1 hour before the addition of lysosphingolipids or CADs. (F) Representative Western blots of P2RX4 and histone H2B (loading control) in lysates of MCF7 cells

transfected with 20 nM non-targeting control (siControl) or 20 nM si*P2RX4* siRNAs for 72 hours. (G-I) Death of MCF7 cells transfected as in (F) and treated with indicated concentrations of GalSph (G), GlcSph (H) or siramesine (I, positive control) for the last 48 hours was determined as in Fig 1A. Error bars, SD of three independent experiments. *, $P < 0.05$; **, $P < 0.01$; ***, $P < 0.001$; ****, $P < 0.0001$ as analyzed by two-way ANOVA followed by Dunnett's (A-E) or Sidak's (G-I) multiple comparisons tests.

line with this, cholesterol had no effect on the decrease of HexSph levels in the culture medium during the first 6 hours after the addition of lysosphingolipids (S2A Fig). Vice versa, the treatment with the lysosphingolipids affected neither the cellular uptake of cholesterol (S2B Fig) nor the levels of cholesterol in the cell culture medium (S2C Fig).

Patients with Niemann-Pick type C disease carry mutations either in NPC intracellular cholesterol transporter 1 or 2 (*NPC1* or *NPC*2) gene, which together are responsible for lysosomal

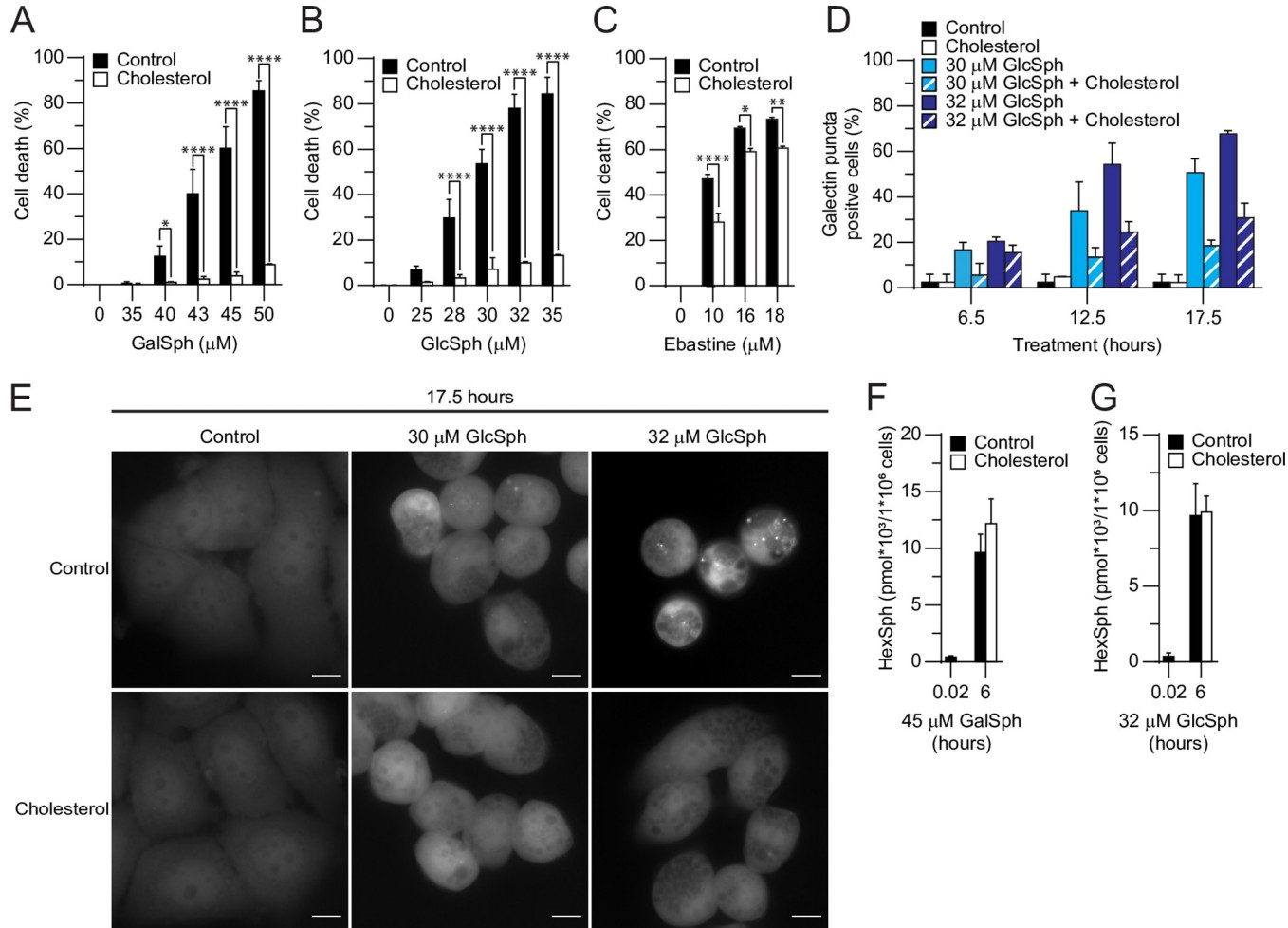

**Fig 6. Cholesterol inhibits cell death induced by GalSph and GlcSph in MCF7 cells.** (A-C) Death of MCF7 cells treated for 48 hours with indicated concentrations of GalSph (A), GlcSph (B), or ebastine (C) was determined as in Fig 1A. When indicated, 60 µM cholesterol was added 24 hours before the addition of lysosphingolipids or CADs. (D) Percentage of galectin-3 puncta positive MCF7 galectin-3-eGFP cells treated as indicated with GlcSph and visualized as in (E). When indicated, 60 µM cholesterol was added 24 hours before the addition of lysosphingolipids (n = 2). (E) Representative images of MCF7 galectin-3-eGFP cells treated with indicated concentrations of GlcSph, for 17.5 hours were acquired using the ImageXpress (*FITC* channel). When indicated, 60 µM cholesterol was added 24 hours before the addition of GlcSph. Scale bars, 10 µm (F-G) Quantities of HexSph (GalSph and GlcSph) in lysates of MCF7 cells treated with 45 µM GalSph (F) and 32 µM GlcSph (G). When indicated, 60 µM cholesterol was added 24 hours before the addition of lysosphingolipids. Error bars, SD of three independent experiments. *, $P < 0.05$; **, $P < 0.01$; ****, $P < 0.0001$ as analyzed by two-way ANOVA followed by Sidak's multiple comparisons tests.

cholesterol efflux [54]. Because the accumulation of cholesterol upon pharmacological inhibition of NPC1 protein has been shown to protect cells from lysosome-dependent cell death [53], we next compared the sensitivity of skin fibroblasts derived from a Niemann-Pick type C patient to those derived from a healthy control. In line with the protective effect of exogenously added cholesterol, fibroblasts from a Niemann-Pick type C patient were significantly more resistant to the cell death induced by GalSph, GlcSph and ebastine than control fibroblasts (Fig 7A–7C).

Notably, the NPC1-associated protection against lysosphingolipids was accompanied by a significant reduction in the treatment-induced lysosomal leakage as defined by the number of galectin-3 puncta-positive cells 6–8 hours after the treatment (Fig 7D–7F).

Taken together, these data strongly suggest that lysosphingolipid-induced lysosomal membrane permeabilization is as at least partially causative of the subsequent cell death.

## Discussion

Here we use MCF7 breast cancer cells as the main model system and provide evidence suggesting that GalSph and GlcSph induce cell death with many similarities to CAD-induced lysosome-dependent cell death. This claim is based on data showing that both lysosphingolipids induce lysosomal membrane permeabilization clearly before the loss of plasma membrane integrity, and that the stabilization of lysosomal membranes with exogenously added cholesterol confers an almost complete protection against lysosphingolipid-induced lysosomal leakage and cell death. Similarities to lysosome-dependent cell death is further emphasized by the lack of caspase activation in cells treated with lethal concentrations of lysosphingolipids, and the inability of well-characterized inhibitors of apoptosis, necroptosis, and ferroptosis to confer protection against lysosphingolipids. Importantly, lysosphingolipids induced lysosomal membrane permeabilization and cell death also in primary fibroblasts, and both events were significantly reduced in Niemann-Pick type C fibroblasts, which are characterized by lysosomal cholesterol accumulation and increased lysosomal membrane stability [53, 54].

We obtained the majority of the presented data using the lysosphingolipids at concentrations of their LC50 values or LC20 and LC50 values. The applied concentrations slightly varied between experiments due to small variations in the potencies between batches of prepared stock solutions of lysosphingolipids. Previous data by Folts and co-workers showing that GalSph and GlcSph increase lysosomal pH and cathepsin activity as a part of their cytotoxicity suggest that GalSph and GlcSph induce a similar lysosomal cell death pathway as CADs [28, 32]. Supporting this idea, we show here that CAD-resistant MCF7 cells are also resistant to the lysosphingolipid-induced cytotoxicity. Additionally, lysosphingolipids activate the cAMP signalling pathway and induce lysosomal leakage as early events occurring prior to cell death, in a manner similar to that observed in CAD-treated cells [22, 28]. The role of cAMP in lysosphingolipid-induced cell death is supported by the ability of forskolin to enhance cell death induced by lysosphingolipids at concentrations below their respective IC50 values. On the contrary, forskolin failed to enhance cell death when combined with higher concentrations of lysosphingolipids at around their respective IC50 values, which were used here to demonstrate the phosphorylation of CREB. In addition, the early lysosphingolipid-induced signalling pathway leading to the cAMP accumulation and cell death in MCF7 cells appears different from the P2RX4- and $Ca^{2+}$-dependent pathway induced by CADs because the depletion of P2RX4 in MCF7 cells had no effect on the lysosphingolipid-induced cytotoxicity. Thus, how lysosphingolipids activate the cAMP signalling pathway and whether it is involved in the cytotoxic mechanism of lysosphingolipids remain to be studied.

Encouraged by the many similarities to CAD-induced lysosome-dependent cell death, we attempted here to rescue MCF7 cells from lysosphingolipid-induced cell death through

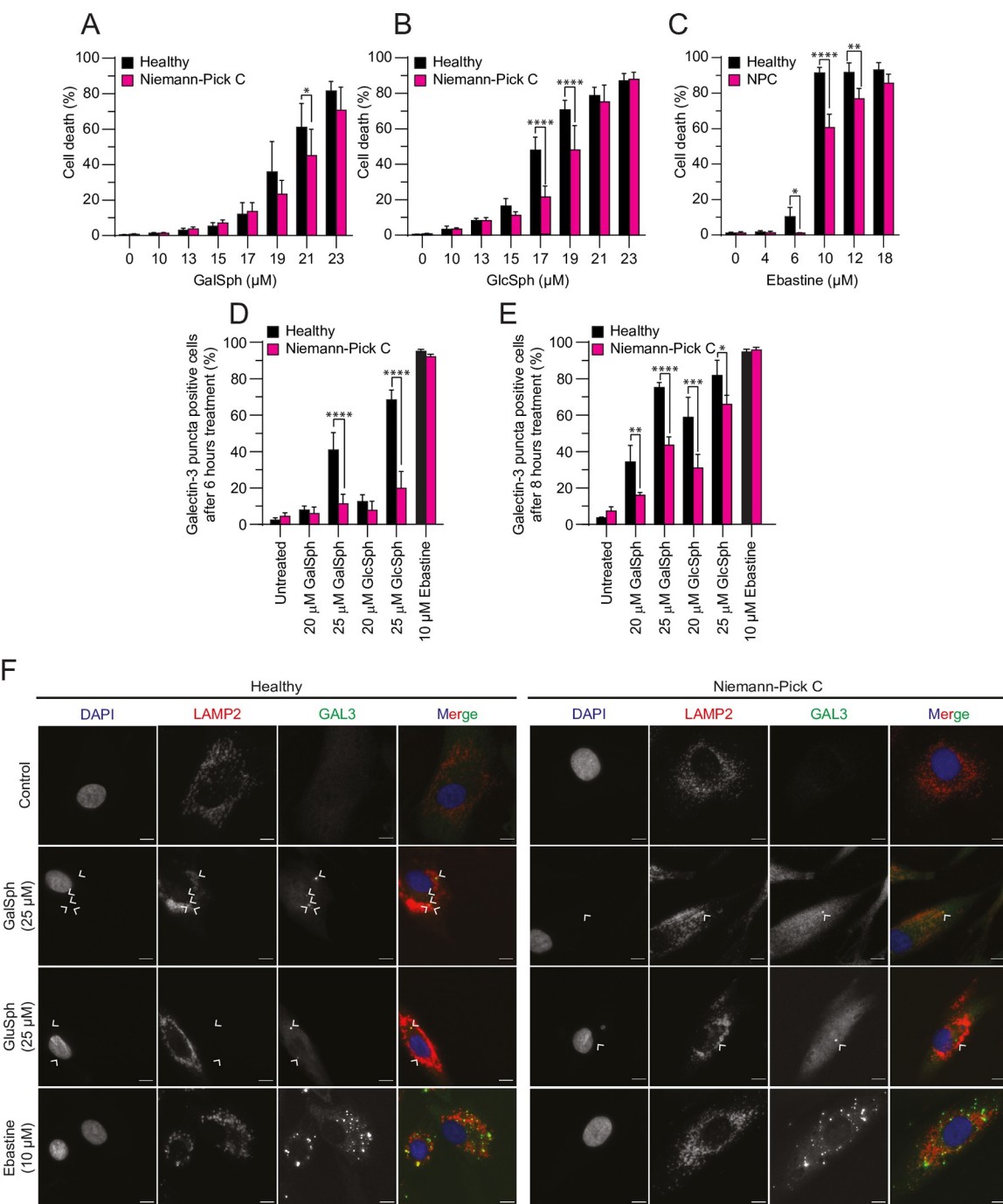

**Fig 7. Niemann-Pick type C fibroblasts are resistant against GalSph- and GlcSph-induced cell death.** (A-C) Death of fibroblasts from healthy controls or patients with Niemann-Pick type C disease (NPC) treated with indicated concentrations of GalSph (A), GlcSph (B), or ebastine (C) for 48 hours was determined as in Fig 1A. (D-E) Percentage of galectin-3 puncta positive fibroblasts from healthy controls or from patients with Niemann-Pick type C disease after 6 hours (D) or 8 hours (E) of indicated treatments was visualized by immunocytochemistry. A minimum of hundred randomly chosen cells was analyzed *per* sample. (F) Representative images of fibroblasts from healthy controls or from patients with Niemann-Pick type C disease treated with indicated concentration of GalSph, or GlcSph, or ebastine for 6 hours were acquired using the ImageXpress (*DAPI, Cy5* and *FITC* channels). Scale bars, 10 μm. Error bars, SD of three independent experiments. *, $P < 0.05$; **, $P < 0.01$; ***, $P < 0.001$; ****, $P < 0.0001$ as analyzed by two-way ANOVA followed Sidak's multiple comparisons tests.

inhibition of cathepsins. Even though we previously demonstrated that a cathepsin inhibitor can partially rescue MCF7 cells from CAD-induced cell death [55], we were unable to demonstrate the same for lysosphingolipid-induced cell death. Lysosphingolipids induce leakage on a small numbers of lysosomes per cells in comparison to ebastine (Fig 3), and we still haven't provided conclusive evidence for the lysosome leakage playing an essential role in the process of lysosphingolipid-induced cell death. Indeed, many other molecular mechanisms of cytotoxicity have been suggested for lysosphingolipids, especially in studies of Gaucher and Krabbe diseases [4, 52, 56–58].

When comparing the lysosphingolipid-induced changes in the lipid profiles of MCF7 cells with those induced by CADs, it is, however different lipid classes that are affected. While the major lysosphingolipid-induced change in the lipidome of MCF7 cells is the decrease in the PC class, treatment of these cells with CADs trigger an early increase in the levels of a broad range of lysoglycerophospholipids classes and a subsequent increase in sphingomyelin levels (Inger Ødum Nielsen, unpublished data) [22]. The degradation of the lysosphingolipids may produce sphingosine, a precursor for sphingolipid biosynthesis, or hexadecanal, which can be converted to palmitic acid [59]. However, the treatment with lysosphingolipids did not increase cellular levels of sphingolipid classes such as Cer, SM, or HexCer, or glycerophospholipid species with incorporated palmitic acid (S6 Table). These data, together with the high HexSph concentrations measured in the lysosphingolipid-treated MCF7 cells suggest that the lysosphingolipids are poorly metabolized within the MCF7 cells.

The present study shows that the treatment of MCF7 breast cancer cells with GalSph or GlcSph both inhibit the growth and induce cell death. We primarily used MCF7 cells here because this cell line has been extensively studied for the CAD-induced cell death. However, the data presented here encourage further studies in other cell lines and on the potential of lysosphingolipids in cancer therapies. Furthermore, our study revealing a novel mechanism of their cytotoxicity through induction of lysosomal leakage encourages possibilities of treating Krabbe and Gaucher diseases with lysosome stabilizing drugs.

## Supporting information

**S1 Fig. GalSph and GlcSph do not induce neutralization of lysosomal pH in MCF7 cells.** The graph indicates measured fluorescein (FITC) and pH-insensitive tetramethylrhodamine (TMR) intensity ratios of vesicles in MCF7 cells treated with vehicle, 10 nM concanamycin A, 53 μM GalSph, or 40 μM GlcSph for 1 or 5 hours. Small circles in light grey symbolizes all data points, large dark grey circles indicate technical replicates, and triangles specify biological replicates colorcoded as individual replicates. *P*-values calculated by unpaired student t-test are shown in the graph. Concanamycin A, a V-ATPase inhibitor, was used as a positive control for lysosomal neutralization.
(TIF)

**S2 Fig. Quantities of cholesterol and HexSph in MCF7 cells.** (A) Quantities of HexSph in cell culture medium of MCF7 cells treated as indicated. (B-C) Quantities of cholesterol in lysates (B) and cell culture medium (C) of MCF7 cells treated as indicated at the time point 6 hours after the addition of GalSph or GlcSph. Error bars, SD of three independent experiments.
(TIF)

**S1 Table. Table of reagents and chemicals.**
(PDF)

**S2 Table. Table of antibodies.**
(PDF)

**S3 Table. Table of siRNA.**
(PDF)

**S4 Table. Internal lipid standards.**
(PDF)

**S5 Table. Precursor ion, fragment ion and neutral loss for lipid identification.**
(PDF)

**S6 Table. Quantities (mol%) of lipid species identified in GalSph- and GlcSph-treated MCF7 cells and their lysosomes.**
(PDF)

**S1 File. Original uncropped and unadjusted images underlying all blots.**
(PDF)

# Acknowledgments

We thank Atul Anand and Signe Diness Vindeløv for preparing the MCF7 CAD resistant cells, Tiina Naumanen Dietrich for outstanding help and introduction to the ImageXpress and MetaXpress analysis, and Dianna Skousborg Larsen and Louise Vanderfox for valuable technical assistance.

# Author Contributions

**Conceptualization:** Kamilla Stahl-Meyer, Thomas Kirkegaard, Nikolaj Havnsøe Torp Petersen, Marja Jäättelä.

**Data curation:** Kamilla Stahl-Meyer, Mesut Bilgin, Jonathan Stahl-Meyer, Kenji Maeda.

**Formal analysis:** Kamilla Stahl-Meyer, Mesut Bilgin, Jonathan Stahl-Meyer, Kenji Maeda.

**Funding acquisition:** Kamilla Stahl-Meyer, Thomas Kirkegaard, Nikolaj Havnsøe Torp Petersen, Kenji Maeda, Marja Jäättelä.

**Investigation:** Kamilla Stahl-Meyer.

**Methodology:** Mesut Bilgin, Jonathan Stahl-Meyer, Kenji Maeda.

**Project administration:** Kamilla Stahl-Meyer, Nikolaj Havnsøe Torp Petersen, Marja Jäättelä.

**Resources:** Thomas Kirkegaard, Nikolaj Havnsøe Torp Petersen, Marja Jäättelä.

**Supervision:** Mesut Bilgin, Thomas Kirkegaard, Nikolaj Havnsøe Torp Petersen, Kenji Maeda, Marja Jäättelä.

**Validation:** Kamilla Stahl-Meyer.

**Visualization:** Kamilla Stahl-Meyer, Kenji Maeda.

**Writing – original draft:** Kamilla Stahl-Meyer, Kenji Maeda, Marja Jäättelä.

**Writing – review & editing:** Kamilla Stahl-Meyer, Lya K. K. Holland, Kenji Maeda, Marja Jäättelä.

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
