## [Decision Letter · Decision Letter 0]

16 May 2022

PONE-D-22-09320Galactosyl- and glucosylsphingosine induce lysosomal membrane permeabilization and lysosome-dependent cell deathPLOS ONE

Dear Dr. Jäättelä,

Thank you for submitting your manuscript to PLOS ONE. After careful consideration, we feel that it has merit but does not fully meet PLOS ONE’s publication criteria as it currently stands. Therefore, we invite you to submit a revised version of the manuscript that addresses the points raised during the review process, which you can find below.

**Reviewer 1**

The manuscript “Galactosyl- and glucosylsphingosine induce lysosomal membrane permeabilization and lysosome-dependent cell death” by Stahl-Meyer et al is well written The presentation is structured and easy to read. The authors have exposed MCF-7 cells to galactosylsphingosine and glucosylsphingosine and identified lysosomal membrane permeabilization before cell death. The mechanism of action is compared to cationic amphiphilic drugs (CADs) which have an established pathway to cause LMP-dependent cell death through activation of cAMP. However the results do not support their conviction. The manuscript should be rewritten with more careful interpretation of the data.

Major comments:

1. The lysosphinolipids are added to cells at different concentrations, which on the one hand contribute to make the study transparent, but also make it very hard to compare results between different panels. For example glucosylsphingosine is used at the concentrations 25, 28, 30, 32, 35, 38, 40, 43, 45 and 50 µM in the presented experiments. From the results in Fig 5 the authors note that glucosylsphingosine may induce cell death through different mechanisms depending on if the concentration is 28 or 32 µM. The results are similar for galactosylsphingosine in a slightly higher concentrations span (different mechanism at 40 and 43 µM). With this very narrow concentration span in mind, it is challenging to interpret and compare the results. The authors need to address this in the discussion.

2. In Figure 3 lysosomal membrane permeabilization is estimated by galectin-3 positive puncta. How many lysosomes must be galectin positive to induce lysosomal dependent cell death? In samples exposed to Ebastine, the CAD that is used as a positive control, galectin-3 puncta are found in 80% of the cells and each cell presents approx. 7 positive puncta. The corresponding numbers for the lysosphinolipids are 45% cells with 2-3 positive puncta/cell. The cell death is, however, much higher in the lysosphinolipids; 80% of the cells has died after 48 h which means that a large proportion of the cells died without showing positive galectine-3 puncta. Considering that Ebastine, although showing high number of galectin puncta, only show 40% cell death after 48h indicates that the MCF-7 cells are not excessively sensitive to broken lysosomes. Thus, the correlation between galectin puncta and cell death is not convincing for lysosphinolipids and the impact of LMP overestimated in the cell death mechanism for the lysosphinolipids.

3. The authors have considered several alternative cell death mechanism with negative result. However, as lysosphinolipids are accumulated in other parts of the cell beside the lysosomes, could the toxicity be caused by mitochondrial damage. Could the toxicity be an effect of increased production of ROS? Would an antioxidant protect?

4. Figure 7: It is surprising that normal and NPC-negative fibroblasts are more sensitive to lysosphinolipids (concentration range 15-23 µM) than MCF-7 cells, while Ebastine toxicity is in the same range in all cell types. Can this be explained?

5. Figure 7. Include galectin puncta results for Ebastine as mentioned in the figure legend 7F. 

Minor:

- Row 425 “…….reached maximal levels already after 2 hours….” According to the graph it should be 4 h.

- Figure 5A Coloring of the bars could be improved to increase the clarity (GalSph and Siramesine are both red and Control and GluSph both black)

**Reviewer 2**

Stahl-Meyer and coworkers show here that GalSph and GlcSph trigger lysosome membrane permeabilization (LMP) and cell death in MCF7 cells. LMP was associated to generation of cAMP and independent of lysosomal calcium release. Moreover, LMP was significantly prevented by cholesterol, a known membrane-stabilizing factor. In keeping with this, fibroblasts from a Niemann-Pick type C patient were protected against GalSph- and GlcSph-induced lysosomal death. The manuscript is interesting and well written. The data are abundant and of high quality; the experimental approach is effective and adequate to answer the biological questions the authors are facing with. Despite the overall appreciation deserved to the research and its results, I have found and listed below a few points that need attention by the Authors.

Major criticisms

1. In my opinion, the reason(s) for using breast cancer cell lines of the luminal A subtype as the main experimental model for this study is not so obvious and not adequately addressed in the manuscript. If the aim of the study is to prove the capability of the lysosphingolipids to specifically trigger kill breast cancer cells, the results should be compared with other generated using normal breast epithelial cells.

2. Page 19, line 308-312. GalSph and GlcSph have an LD50 of 44 and 33 µM, respectively. Fig S1 shows that the number of MCF7 cells is markedly reduced by GalSph and GlcSph already at 40 and 30 µM, and this effect is claimed to be produced by using a ‘sublethal’ concentration of sphingolipids. However, Fig. 1A shows that these concentrations already trigger cell death, which thus cannot be considered ‘sublethal’. In addition, the concepts of inhibition of cell growth and of induction of cell death are used as synonyms, which is not correct. The paragraph needs revision.

3. Page 21, lines 340-342. Although the data on LMP induction are clear and basically convincing, occurrence of lysosomal death should be further confirmed by the demonstration that cathepsin inhibitors or elevation of intracellular pH prevent cell death.

4. Page 25, about the effect of cholesterol on lysosphingolipid-induced cell death. Have the Authors verified whether cholesterol supplementation affects CAD- or lysosphingolipid-induced cAMP generation?

5. Page 29, lines 553-555. The data presented in the manuscript refer to MCF7 cells only, one among many models of the multiple breast cancer molecular subtypes. Thus, although in my opinion it is likely to be true, this sentence is not adequately supported by direct experimental evidences, thus should be modified, unless better validated in additional cancer models.

Minor criticisms

- Page 13, line 160. ‘Greisner…’ should be probably changed to ‘Greiner…’. Please check.

- Page 16, line 221. ‘+’ should be changed to ‘±’.

- Page 21, lines 351-353. The aim of this experimental dataset is to verify that both Gal- and GlcSph enter the cells. Thus, I would have expected a sentence stating this fact, not pointing out the decrease of HexSph in the medium (the complementary information) which, on the other hand, does not unambiguously proves their uptake by the cells. Although I understand that this comment refers to the one’s feeling and for this is basically questionable, should the Authors agree with this view the sentence could be updated to evidence the uptake and the intracellular accumulation of lysosphingolipids.

**Reviewer 3**

In the present study, Dr. Stahl-Meyer K et al. investigated the effects of lysosphingolipids, galactosylsphingosine (GalSph) and glucosylsphingosine (GlcSph), on lysosomes in comparison to lysosome-destabilizing cationic amphiphilic drugs (CADs). The authors showed that GalSph and GlcSph induce lysosomal membrane permeabilization and lysosome-dependent cell death in MCF-7 breast cancer cells and human fibroblasts. The authors also showed that the contribution of cAMP and Ca2+ pathways in inducing cell death was different in lysosphingolipids and CADs. Since GalSph and GlcSph are lysophospholipids that accumulate in the brains of patients with Krabbe disease and Gaucher disease, respectively, this study deals with an important theme and may be of interest to wide-readers. The data presented in this manuscript roughly support the authors’ conclusion, but the authors need to address the following comments to confirm their conclusions.

Specific comments:

1. Figure 3: The enhancement of lysosomal permeability by GalSph (50 or 53 µM) or GlcSph (38 or 40 µM) is weaker than that of ebastine (12 or 16 µM). On the other hand, the induction of cell death by GalSph or GlcSph is stronger than that of ebastine. Taken together, it is possible to consider the contribution of the enhancement of lysosomal permeability to the induction of cell death by GalSph and GlcSph is small. At least there should be some discussion about this contradiction.

2. Figure 5: It is desirable to analyze intracellular cAMP levels and Ca2+ efflux, as the authors did in a previous paper. In addition, it is considered that the involvement of cAMP can be clarified by using lysophospholipids in combination with a reagent that lowers cAMP rather than in combination with forskolin that raises cAMP. Similarly, it is desirable to analyze the lysosomal pH.

3. Figure 6F & G: Although the authors are measuring intracellular GalSph and GlcSph, it is desirable to measure GalSph and GlcSph in lysosomes, as is done in Figure 4.

4. Figure 1E-G: It is desirable to have positive controls for necrostatin-1 and ferrostatin-1.

5. P6L117: The authors describe fibroblasts from patients with Gaucher's disease in the Materials and Methods section but have not shown any results using these cells.

6. P19L456-462: The instructions in Figures 6D and 6E are reversed.

We look forward to receiving your revised manuscript.

Kind regards,

Vladimir Trajkovic

Academic Editor

PLOS ONE

Journal Requirements:

[This work was supported by grants from the Danish Cancer Society (R269-A15695), Danish National Research Foundation (DNRF125) and NovoNordisk Foundation (NNF19OC0054296) to M.J., and by the Independent Research Fund Denmark (6108–00542B) and Novo Nordisk Foundation (NNF17OC0029432) to K.M., and by Innovation Fund Denmark (7038-00062B) to K.SM., N.H.T.P., and T.K.]

 [This work was supported by grants from the Danish Cancer Society (R269-A15695), Danish National Research Foundation (DNRF125) and NovoNordisk Foundation (NNF19OC0054296) to M.J. and by the Independent Research Fund Denmark (6108–00542B) and Novo Nordisk Foundation (NNF17OC0029432) to K.M., and and by Innovation Fund Denmark (7038-00062B) to K.SM., N.H.T.P., and T.K.]

Reviewers' comments:

Reviewer's Responses to Questions

**Comments to the Author**

1. Is the manuscript technically sound, and do the data support the conclusions?

Reviewer #1: Partly

Reviewer #2: Yes

Reviewer #3: Yes

2. Has the statistical analysis been performed appropriately and rigorously? 

Reviewer #1: Yes

Reviewer #2: Yes

Reviewer #3: Yes

3. Have the authors made all data underlying the findings in their manuscript fully available?

Reviewer #1: Yes

Reviewer #2: Yes

Reviewer #3: Yes

4. Is the manuscript presented in an intelligible fashion and written in standard English?

Reviewer #1: Yes

Reviewer #2: Yes

Reviewer #3: Yes

5. Review Comments to the Author

Reviewer #1: The manuscript “Galactosyl- and glucosylsphingosine induce lysosomal membrane permeabilization and lysosome-dependent cell death” by Stahl-Meyer et al is well written The presentation is structured and easy to read. The authors have exposed MCF-7 cells to galactosylsphingosine and glucosylsphingosine and identified lysosomal membrane permeabilization before cell death. The mechanism of action is compared to cationic amphiphilic drugs (CADs) which have an established pathway to cause LMP-dependent cell death through activation of cAMP. However the results do not support their conviction. The manuscript should be rewritten with more careful interpretation of the data.

Major comments:

The lysosphinolipids are added to cells at different concentrations, which on the one hand contribute to make the study transparent, but also make it very hard to compare results between different panels.

For example glucosylsphingosine is used at the concentrations 25, 28, 30, 32, 35, 38, 40, 43, 45 and 50 µM in the presented experiments. From the results in Fig 5 the authors note that glucosylsphingosine may induce cell death through different mechanisms depending on if the concentration is 28 or 32 µM. The results are similar for galactosylsphingosine in a slightly higher concentrations span (different mechanism at 40 and 43 µM). With this very narrow concentration span in mind, it is challenging to interpret and compare the results. The authors need to address this in the discussion.

In Figure 3 lysosomal membrane permeabilization is estimated by galectin-3 positive puncta. How many lysosomes must be galectin positive to induce lysosomal dependent cell death? In samples exposed to Ebastine, the CAD that is used as a positive control, galectin-3 puncta are found in 80% of the cells and each cell presents approx. 7 positive puncta. The corresponding numbers for the lysosphinolipids are 45% cells with 2-3 positive puncta/cell. The cell death is, however, much higher in the lysosphinolipids; 80% of the cells has died after 48 h which means that a large proportion of the cells died without showing positive galectine-3 puncta. Considering that Ebastine, although showing high number of galectin puncta, only show 40% cell death after 48h indicates that the MCF-7 cells are not excessively sensitive to broken lysosomes. Thus, the correlation between galectin puncta and cell death is not convincing for lysosphinolipids and the impact of LMP overestimated in the cell death mechanism for the lysosphinolipids.

The authors have considered several alternative cell death mechanism with negative result. However, as lysosphinolipids are accumulated in other parts of the cell beside the lysosomes, could the toxicity be caused by mitochondrial damage. Could the toxicity be an effect of increased production of ROS? Would an antioxidant protect?

Figure 7: It is surprising that normal and NPC-negative fibroblasts are more sensitive to lysosphinolipids (concentration range 15-23 µM) than MCF-7 cells, while Ebastine toxicity is in the same range in all cell types. Can this be explained?

Figure 7. Include galectin puncta results for Ebastine as mentioned in the figure legend 7F.

Minor:

Row 425 “…….reached maximal levels already after 2 hours….” According to the graph it should be 4 h.

Figure 5A Coloring of the bars could be improved to increase the clarity (GalSph and Siramesine are both red and Control and GluSph both black)

Reviewer #2: Stahl-Meyer and coworkers show here that GalSph and GlcSph trigger lysosome membrane permeabilization (LMP) and cell death in MCF7 cells. LMP was associated to generation of cAMP and independent of lysosomal calcium release. Moreover, LMP was significantly prevented by cholesterol, a known membrane-stabilizing factor. In keeping with this, fibroblasts from a Niemann-Pick type C patient were protected against GalSph- and GlcSph-induced lysosomal death.

The manuscript is interesting and well written. The data are abundant and of high quality; the experimental approach is effective and adequate to answer the biological questions the authors are facing with. Despite the overall appreciation deserved to the research and its results, I have found and listed below a few points that need attention by the Authors.

Major criticisms

In my opinion, the reason(s) for using breast cancer cell lines of the luminal A subtype as the main experimental model for this study is not so obvious and not adequately addressed in the manuscript. If the aim of the study is to prove the capability of the lysosphingolipids to specifically trigger kill breast cancer cells, the results should be compared with other generated using normal breast epithelial cells.

Page 19, line 308-312. GalSph and GlcSph have an LD50 of 44 and 33 µM, respectively. Fig S1 shows that the number of MCF7 cells is markedly reduced by GalSph and GlcSph already at 40 and 30 µM, and this effect is claimed to be produced by using a ‘sublethal’ concentration of sphingolipids. However, Fig. 1A shows that these concentrations already trigger cell death, which thus cannot be considered ‘sublethal’. In addition, the concepts of inhibition of cell growth and of induction of cell death are used as synonyms, which is not correct. The paragraph needs revision.

Page 21, lines 340-342. Although the data on LMP induction are clear and basically convincing, occurrence of lysosomal death should be further confirmed by the demonstration that cathepsin inhibitors or elevation of intracellular pH prevent cell death.

Page 25, about the effect of cholesterol on lysosphingolipid-induced cell death. Have the Authors verified whether cholesterol supplementation affects CAD- or lysosphingolipid-induced cAMP generation?

Page 29, lines 553-555. The data presented in the manuscript refer to MCF7 cells only, one among many models of the multiple breast cancer molecular subtypes. Thus, although in my opinion it is likely to be true, this sentence is not adequately supported by direct experimental evidences, thus should be modified, unless better validated in additional cancer models.

Minor criticisms

Page 13, line 160. ‘Greisner…’ should be probably changed to ‘Greiner…’. Please check.

Page 16, line 221. ‘+’ should be changed to ‘±’.

Page 21, lines 351-353. The aim of this experimental dataset is to verify that both Gal- and GlcSph enter the cells. Thus, I would have expected a sentence stating this fact, not pointing out the decrease of HexSph in the medium (the complementary information) which, on the other hand, does not unambiguously proves their uptake by the cells. Although I understand that this comment refers to the one’s feeling and for this is basically questionable, should the Authors agree with this view the sentence could be updated to evidence the uptake and the intracellular accumulation of lysosphingolipids.

Reviewer #3: In the present study, Dr. Stahl-Meyer K et al. investigated the effects of lysosphingolipids, galactosylsphingosine (GalSph) and glucosylsphingosine (GlcSph), on lysosomes in comparison to lysosome-destabilizing cationic amphiphilic drugs (CADs). The authors showed that GalSph and GlcSph induce lysosomal membrane permeabilization and lysosome-dependent cell death in MCF-7 breast cancer cells and human fibroblasts. The authors also showed that the contribution of cAMP and Ca2+ pathways in inducing cell death was different in lysosphingolipids and CADs. Since GalSph and GlcSph are lysophospholipids that accumulate in the brains of patients with Krabbe disease and Gaucher disease, respectively, this study deals with an important theme and may be of interest to wide-readers. The data presented in this manuscript roughly support the authors’ conclusion, but the authors need to address the following comments to confirm their conclusions.

Specific comments:

1. Figure 3: The enhancement of lysosomal permeability by GalSph (50 or 53 µM) or GlcSph (38 or 40 µM) is weaker than that of ebastine (12 or 16 µM). On the other hand, the induction of cell death by GalSph or GlcSph is stronger than that of ebastine. Taken together, it is possible to consider the contribution of the enhancement of lysosomal permeability to the induction of cell death by GalSph and GlcSph is small. At least there should be some discussion about this contradiction.

2. Figure 5: It is desirable to analyze intracellular cAMP levels and Ca2+ efflux, as the authors did in a previous paper. In addition, it is considered that the involvement of cAMP can be clarified by using lysophospholipids in combination with a reagent that lowers cAMP rather than in combination with forskolin that raises cAMP. Similarly, it is desirable to analyze the lysosomal pH.

3. Figure 6F & G: Although the authors are measuring intracellular GalSph and GlcSph, it is desirable to measure GalSph and GlcSph in lysosomes, as is done in Figure 4.

4. Figure 1E-G: It is desirable to have positive controls for necrostatin-1 and ferrostatin-1.

5. P6L117: The authors describe fibroblasts from patients with Gaucher's disease in the Materials and Methods section but have not shown any results using these cells.

6. P19L456-462: The instructions in Figures 6D and 6E are reversed.

6. PLOS authors have the option to publish the peer review history of their article (what does this mean?). If published, this will include your full peer review and any attached files.

Reviewer #1: No

Reviewer #2: No

Reviewer #3: No

---

## [Author Response · Author response to Decision Letter 0]

29 Jun 2022

Reviewer 1

The manuscript “Galactosyl- and glucosylsphingosine induce lysosomal membrane permeabilization and lysosome-dependent cell death” by Stahl-Meyer et al is well written. The presentation is structured and easy to read. The authors have exposed MCF-7 cells to galactosylsphingosine and glucosylsphingosine and identified lysosomal membrane permeabilization before cell death. The mechanism of action is compared to cationic 

amphiphilic drugs (CADs) which have an established pathway to cause LMP-dependent cell death through activation of cAMP. However the results do not support their conviction. The manuscript should be rewritten with more careful interpretation of the data.

Major comments:

1. The lysosphinolipids are added to cells at different concentrations, which on the one hand contribute to make the study transparent, but also make it very hard to compare results between different panels. For example glucosylsphingosine is used at the concentrations 25, 28, 30, 32, 35, 38, 40, 43, 45 and 50 µM in the presented experiments. From the results in Fig 5 the authors note that glucosylsphingosine may induce cell death through different mechanisms depending on if the concentration is 28 or 32 µM. The results are similar for galactosylsphingosine in a slightly higher concentrations span (different mechanism at 40 and 43 µM). With this very narrow concentration span in mind, it is challenging to interpret and compare the results. The authors need to address this in the discussion.

We thank the reviewers for the comments. We essentially used the concentrations of LC50 and LC20 values for the comparisons of the effects of lysosphingolipids and cationic amphiphilic drugs (CADs). The values were different between the two lysosphingolipids, and also varied slightly 

between some of the performed experiments, as we determined the LC50 and LC20 values for each batch of the lysosphingolipids we purchased and prepared. 

We have clarified this issue in the discussion section of the revised manuscript. We inserted the following sentence in lines 563-566, page 24 in the revised manuscript with track changes; “We obtained the majority of the presented data using the lysosphingolipids at concentrations of their 

LC50 values or LC20 and LC50 values. The applied concentrations slightly varied between experiments due to small variations in the potencies between batches of prepared stock solutions of lysosphingolipids.” 

2. In Figure 3 lysosomal membrane permeabilization is estimated by galectin-3 positive puncta. How many lysosomes must be galectin positive to induce lysosomal dependent cell death? In samples exposed to Ebastine, the CAD that is used as a positive control, galectin-3 puncta are 

found in 80% of the cells and each cell presents approx. 7 positive puncta. The corresponding numbers for the lysosphinolipids are 45% cells with 2-3 positive puncta/cell. The cell death is, however, much higher in the lysosphinolipids; 80% of the cells has died after 48 h which means 

that a large proportion of the cells died without showing positive galectine-3 puncta. Considering that Ebastine, although showing high number of galectin puncta, only show 40% cell death after 48h indicates that the MCF-7 cells are not excessively sensitive to broken 

lysosomes. Thus, the correlation between galectin puncta and cell death is not convincing for lysosphingolipids and the impact of LMP overestimated in the cell death mechanism for the lysosphingolipids.

There is not necessarily a direct correlation between the number of leaky lysosomes and cell death. For example, treatment of cells with LLOMe can induce leakage of nearly all lysosomes without causing cell death. This discrepancy has been explained by data showing that lysosomal 

membranes become first permeable to molecules with a mass < 10 kDa resulting in the loss of lysosomal acidity and inactivation of cathepsins before the “holes” in the membrane grow and allow the leakage of cathepsins and entrance of galectins (DOI: 10.1242/jcs.204529). It should 

also be noted that both CADs and lipids used in this study don’t induce a complete rupture of lysosomes, but only a leakage of a small portion of their hydrolases. Thus, the number of galectin positive lysosomes does not necessarily correlate with the amount of released cathepsins. 

Nevertheless, we agree that our data do not fully confirm the importance of the lysosomal leakage for the lysosphingolipid-induced cell death. We have added a short paragraph discussing this issue in lines 580-589, pages 24-25 of the discussion section of the revised manuscript with track changes; “Encouraged by the many similarities to CAD-induced lysosome-dependent cell death, we attempted here to rescue MCF7 cells from lysosphingolipid-induced cell death through inhibition of cathepsins. Even though we previously demonstrated that a cathepsin inhibitor can partially rescue MCF7 cells from CAD-induced cell death (55), we were unable to demonstrate the same for lysosphingolipid-induced cell death. Lysosphingolipids induce leakage on a small numbers of lysosomes per cells in comparison to ebastine (Fig 3), and we still haven’t provided conclusive evidence for the lysosome leakage playing an essential role in the process of lysosphingolipid induced cell death. Indeed, many other molecular mechanisms of cytotoxicity have been suggested for lysosphingolipids, especially in studies of Gaucher and Krabbe diseases (4, 52, 56-58).”

3. The authors have considered several alternative cell death mechanism with negative result. However, as lysosphingolipids are accumulated in other parts of the cell beside the lysosomes, could the toxicity be caused by mitochondrial damage. Could the toxicity be an effect of increased production of ROS? Would an antioxidant protect?

Our data do not exclude the possibility that these lysosphingolipids induce alternative signaling pathways of cell death in addition to those directly linked to their effects on lysosomes. Increased ROS production is indeed suggested as a possible molecular mechanism of Gaucher and Krabbe diseases and our lipidomics data also demonstrate that extracellularly supplied lysosphingolipids are not exclusively present in the lysosomes. 

4. Figure 7: It is surprising that normal and NPC-negative fibroblasts are more sensitive to lysosphinolipids (concentration range 15-23 µM) than MCF-7 cells, while Ebastine toxicity is in the same range in all cell types. Can this be explained?

We do not have an explanation for this. However, CADs and lysosphingolipids are certainly different in both their manners of cell uptake and metabolism. CADs diffuse into cells, but lysosphingolipids with the large polar head group are more likely to enter the cells mainly via 

endocytosis. Lysosphingolipids can also be metabolized and degraded by cellular enzymes such as (glucosylceramidase (GBA1) and galactosylceramidase (GALC). CADs are also degraded slowly in cancer cells (unpublished data) but most likely not by GBA1 or GALC. We speculate that these differences could be reasons that some cell lines can be more sensitive to lysosphingolipids without at the same time being more sensitive to CADs. And, despite many similarities in cell death induced by lysosphingolipids and CADs, we show that they use different signaling pathways to induce lysosomal membrane permeabilization. 

5. Figure 7. Include galectin puncta results for Ebastine as mentioned in the figure legend 7F. 

Thanks to pointing this out. In the originally submitted manuscript, we removed the images of puncta in ebastine-treated cells to allow more space for the images of lysosphingolipid-treated ones, but did not properly remove the figure legend. As suggested, we have included the suggested images and quantifications into the revised figure. 

Minor:

- Row 425 “…….reached maximal levels already after 2 hours….” According to the graph it should be 4 h.

We agree and corrected the manuscript in line 451, page 19 in the revised manuscript with track changes.

- Figure 5A Coloring of the bars could be improved to increase the clarity (GalSph and Siramesine are both red and Control and GluSph both black)

We agree. We have changed the colors in the graph of Fig 5A as suggested. 

Reviewer 2

Stahl-Meyer and coworkers show here that GalSph and GlcSph trigger lysosome membrane permeabilization (LMP) and cell death in MCF7 cells. LMP was associated to generation of cAMP and independent of lysosomal calcium release. Moreover, LMP was significantly prevented by cholesterol, a known membrane-stabilizing factor. In keeping with this, fibroblasts from a Niemann-Pick type C patient were protected against GalSph- and GlcSph-induced lysosomal death. The manuscript is interesting and well written. The data are abundant and of high quality; the experimental approach is effective and adequate to answer the biological questions the authors are facing with. Despite the overall appreciation deserved to the research and its results, I have found and listed below a few points that need attention by the Authors.

Major criticisms

1. In my opinion, the reason(s) for using breast cancer cell lines of the luminal A subtype as the main experimental model for this study is not so obvious and not adequately addressed in the manuscript. If the aim of the study is to prove the capability of the lysosphingolipids to specifically trigger kill breast cancer cells, the results should be compared with other generated using normal breast epithelial cells.

We thank the reviewer for the comments. Our intension with the present study was not to perform a comparison between cancer cells vs normal cells, but to demonstrate the effects of lysosphingolipids on the lysosomes and consequences of them. In that context, we intended to illuminate the effects of lysosphingolipids in comparison to those of cationic amphiphilic drugs (CADs), well studied for inducing lysosomal leakage and lysosome-dependent cell death. We have mainly used MCF7 cells in this study, since the effects of CADs are particularly well studied in MCF7 

cells and because experimental tools, setups, and protocols are also best established for these cells. We have highlighted this in the revised manuscript with track changes, lines 603-604, page 25 with the following sentence; “We primarily used MCF7 cells here because this cell line has been extensively studied for the CAD-induced cell death.”

2. Page 19, line 308-312. GalSph and GlcSph have an LD50 of 44 and 33 µM, respectively. Fig S1 shows that the number of MCF7 cells is markedly reduced by GalSph and GlcSph already at 40 and 30 µM, and this effect is claimed to be produced by using a ‘sublethal’ concentration of sphingolipids. However, Fig. 1A shows that these concentrations already trigger cell death, which thus cannot be considered ‘sublethal’. In addition, the concepts of inhibition of cell growth and of induction of cell death are used as synonyms, which is not correct. The paragraph 

needs revision.

We agree that the concentrations of lysosphingolipids used in Fig S1 of the original manuscript to examine the growth inhibitory effects are inducing cell death and thus not sublethal, according to cell death assay shown in Fig 1A. The problem here is the small shift in the potencies of the lysosphingolipids between batches of prepared stock solutions and thus between some of the presented experiments. The data of growth inhibition presented in Fig 1S come from the same readouts/measurements as those of cell death in Fig 2A (Fig S1 counting total cell numbers and Fig 2A the numbers of dead cells/numbers of total cells). In this experiment (of Fig 2A and S1), performed three times independently, the LC50 values of parental MCF7 cells for Gal- and GlcSph were slightly higher at around 50-53 µM and 38-40 µM in comparison to 44 and 33 µM, 

respectively, determined in the experimented presented in Fig 1A. To eliminate this confusing manner of data representation, we have merged Fig 2 with Fig S1 (to new Fig 2), so that the data on cell death and cell growth can be compared next to each other. Additionally, we have also added a short paragraph in the discussion section describing the issue of batch variation in the potency of lysosphingolipids in page 24, lines 564-566 in the revised manuscript with track changes; “The applied concentrations slightly varied between experiments due to small variations in the potencies between batches of prepared stock solutions of lysosphingolipids.”

3. Page 21, lines 340-342. Although the data on LMP induction are clear and basically convincing, occurrence of lysosomal death should be further confirmed by the demonstration that cathepsin inhibitors or elevation of intracellular pH prevent cell death.

We agree that the induction of lysosome-dependent cell death has only weakly been demonstrated in the originally submitted manuscript. As suggested, we have performed an additional experiment, in which we treated MCF7 cells with cathepsin inhibitors zFA-fmk or Ca-O74-Me, or

with a V-ATPase inhibitor concanamycin A (to inhibit the maturation of cathepsins and thereby reduce cellular cathepsin activity) together with lysosphingolipids. These pre-treatments resulted, however, increased cell death (see below). The known cytotoxicity of the inhibitors might have 

exceeded their possible rescuing effects and concanamycin A might have interfered with the lysosomal degradation of lysosphingolipids. 

As described in the cover letter, we have in the revised manuscript therefore soften our statement regarding the induction of lysosome-dependent cell death by the lysosphingolipids. We have inserted a small paragraph in 580-589, pages 24-25 of the revised manuscript with track changes, discussing the above described results; “Encouraged by the many similarities to CAD-induced lysosome-dependent cell death, we attempted here to rescue MCF7 cells from lysosphingolipid-induced cell death through inhibition of cathepsins. Even though we previously demonstrated that a cathepsin inhibitor can partially rescue MCF7 cells from CAD-induced cell death (55), we were unable to demonstrate the same for lysosphingolipid-induced cell death. Lysosphingolipids induce leakage on a small numbers of lysosomes per cells in comparison to ebastine (Fig 3), and we still haven’t provided conclusive evidence for the lysosome leakage playing an essential role in the process of lysosphingolipid-induced cell death. Indeed, many other molecular mechanisms of cytotoxicity have been suggested for lysosphingolipids, especially in studies of Gaucher and Krabbe diseases (4, 52, 56-58).”

4. Page 25, about the effect of cholesterol on lysosphingolipid-induced cell death. Have the Authors verified whether cholesterol supplementation affects CAD- or lysosphingolipid-induced cAMP generation?

No, we have not tried that yet in this study nor in any previous studies in our laboratory. Addressing this is beyond the scope of the manuscript due to time limitation, but we have currently an ongoing study in which we more thoroughly examine the role of cholesterol on lysosomal 

stability and functioning.

5. Page 29, lines 553-555. The data presented in the manuscript refer to MCF7 cells only, one among many models of the multiple breast cancer molecular subtypes. Thus, although in my opinion it is likely to be true, this sentence is not adequately supported by direct experimental 

evidences, thus should be modified, unless better validated in additional cancer models.

We agree. We modified the statement to: “However, the data presented here encourage further studies in other cell lines and on the potential of lysosphingolipids in cancer therapies” at lines 605-606, page 25 in the revised manuscript with track changes.

Minor criticisms

- Page 13, line 160. ‘Greisner…’ should be probably changed to ‘Greiner…’. Please check.

Yes. We have corrected this at line 162, page 7 in the revised manuscript with track changes.

- Page 16, line 221. ‘+’ should be changed to ‘±’.

Yes. We have corrected this at line 238, page 11 in the revised manuscript with track changes.

- Page 21, lines 351-353. The aim of this experimental dataset is to verify that both Gal- and 

GlcSph enter the cells. Thus, I would have expected a sentence stating this fact, not pointing out the decrease of HexSph in the medium (the complementary information) which, on the other hand, does not unambiguously proves their uptake by the cells. Although I understand that this 

comment refers to the one’s feeling and for this is basically questionable, should the Authors agree with this view the sentence could be updated to evidence the uptake and the intracellular accumulation of lysosphingolipids.

We have actually described this in the originally submitted manuscript, which now is in lines 397-398 in the revised manuscript with track changes; “In parallel, the level of HexSph in the harvested cells increased rapidly during the first hour of treatments with GalSph or GlcSph and continued to increase for 24 hours (Figs 4A and B).” Unfortunately, this sentence is, in the current format of the text, separated from the immediately preceding sentence describing the decrease of lysosphingolipid concentrations in the medium with a figure legend. This might be the reason that the sentence was missed.

Reviewer 3

In the present study, Dr. Stahl-Meyer K et al. investigated the effects of lysosphingolipids, galactosylsphingosine (GalSph) and glucosylsphingosine (GlcSph), on lysosomes in comparison to lysosome-destabilizing cationic amphiphilic drugs (CADs). The authors showed that GalSph and GlcSph induce lysosomal membrane permeabilization and lysosome-dependent cell death in MCF-7 breast cancer cells and human fibroblasts. The authors also showed that the contribution of cAMP and Ca2+ pathways in inducing cell death was different in 

lysosphingolipids and CADs. Since GalSph and GlcSph are lysophospholipids that accumulate in the brains of patients with Krabbe disease and Gaucher disease, respectively, this study deals with an important theme and may be of interest to wide-readers. The data presented in this 

manuscript roughly support the authors’ conclusion, but the authors need to address the following comments to confirm their conclusions.

Specific comments:

1. Figure 3: The enhancement of lysosomal permeability by GalSph (50 or 53 µM) or GlcSph (38 or 40 µM) is weaker than that of ebastine (12 or 16 µM). On the other hand, the induction of cell death by GalSph or GlcSph is stronger than that of ebastine. Taken together, it is possible to 

consider the contribution of the enhancement of lysosomal permeability to the induction of cell death by GalSph and GlcSph is small. At least there should be some discussion about this contradiction.

We thank the reviewer for the comments. The extent of lysosomal leakage assessed with galectin puncta assay might not fully reflect the true extent of leakage of cytotoxic compounds from lysosomes, but we agree that we have not provided fully conclusive evidence for that the lysosomal leakage is an essential event leading to the lysosphingolipid-induced cell death. We have inserted a small paragraph in the discussion section, lines 580-589, pages 24-25 in the revised manuscript with track changes, discussing this issue and the results of the additionally performed experiments: Encouraged by the many similarities to CAD-induced lysosome-dependent cell death, we attempted here to rescue MCF7 cells from lysosphingolipid-induced cell death through inhibition of cathepsins. Even though we previously demonstrated that a cathepsin inhibitor can partially rescue MCF7 cells from CAD-induced cell death (55), we were unable to demonstrate the same for lysosphingolipid-induced cell death. Lysosphingolipids induce leakage on a small numbers of lysosomes per cells in comparison to ebastine (Fig 3), and we still haven’t provided conclusive evidence for the lysosome leakage playing an essential role in the process of lysosphingolipid-induced cell death. Indeed, many other molecular mechanisms of cytotoxicity have been suggested for lysosphingolipids, especially in studies of Gaucher and Krabbe diseases (4, 52, 56-58).”

2. Figure 5: It is desirable to analyze intracellular cAMP levels and Ca2+ efflux, as the authors did in a previous paper. In addition, it is considered that the involvement of cAMP can be clarified by using lysophospholipids in combination with a reagent that lowers cAMP rather 

than in combination with forskolin that raises cAMP. Similarly, it is desirable to analyze the lysosomal pH.

We have previously measured Ca2+ release after treatment with the lysosphingolipids, but the result has been inconclusive. We have also shown in the originally submitted manuscript that the knockdown of P2RX4 did not have any impact on the lysosphingolipid-induced cell death, and we

therefore believe that lysosphingolipids does not induce rapid Ca2+ release like CADs. Our data suggest that lysosphingolipids and CADs induce similar manners of cell death, but they don’t induce cell death or lysosomal membrane permeabilization through the exact same mechanisms. 

We have performed pH measurements, as suggested, to examine whether lysosphingolipids also neutralize lysosomes similarly to CADs prior to lysosomal membrane permeabilization. The lysosphingolipids did not neutralize the lysosome pH after one or five hours of treatments, despite 

the lysosphingolipids being weak bases like CADs. This new data is presented as a supplementary figure, S1 Fig and the results are described in lines 361-365, page 16: “Unlike CADs (50), the lysosphingolipids did not induce notable elevation in the lysosomal pH in MCF7 cells after one or 

five hours of treatments and thus prior to lysosomal leakage, when assessed after loading lysosomes with dextran coupled with pH-sensitive fluorescein and pH-insensitive tetramethylrhodamine (S1 Fig).”

3. Figure 6F & G: Although the authors are measuring intracellular GalSph and GlcSph, it is desirable to measure GalSph and GlcSph in lysosomes, as is done in Figure 4.

We agree that the GalSph and GlcSph levels in the lysosomes are not necessarily fully reflected in the quantification in the cell lysates. We however are currently not able to purify lysosomes from the cholesterol-fed cells, due to the incompatibility of the purification method. We purify lysosomes by feeding cells with iron-dextran particles. These particles very easily aggregate in general to obscure the lysosome purification (as they stick to everything), and the presence of cholesterol in the medium enhances their aggregation. 

4. Figure 1E-G: It is desirable to have positive controls for necrostatin-1 and ferrostatin-1. 

We agree a positive control would have been ideal. We indeed here used an apoptosis inhibitor in combination with an inducer. We however chose to not use positive controls for necrosis and ferroptosis in part because of the challenge in inducing clear-cut necrosis or ferroptosis. MCF7 cells are previously demonstrated to be resistance to ferroptosis induced by a variety of inducers (DOI: 10.3390/antiox11020298), to make the use of a positive control in combination with the inhibitor a challenge (also makes it not so likely that they undergo ferroptosis). For necrostatin-1, we have previously demonstrated its inhibitory effect in MCF7 cells using starvation as an inducer of cell death (DOI:10.1074/jbc.M111.269134)

5. P6L117: The authors describe fibroblasts from patients with Gaucher's disease in the Materials and Methods section but have not shown any results using these cells.

This is correct and these lines are deleted from the Materials and Methods section of the revised 

manuscript.

6. P19L456-462: The instructions in Figures 6D and 6E are reversed.

Yes. We have corrected this in the revised manuscript.

---

## [Decision Letter · Decision Letter 1]

1 Aug 2022

PONE-D-22-09320R1Galactosyl- and glucosylsphingosine induce lysosomal membrane permeabilization and cell death in cancer cellsPLOS ONE

Dear Dr. Jäättelä,

Thank you for submitting your manuscript to PLOS ONE. After careful consideration, we feel that it has merit but does not fully meet PLOS ONE’s publication criteria as it currently stands. Therefore, we invite you to submit a revised version of the manuscript that addresses the points raised during the review process.

Reviewer 3

In the revised version, the authors have done some additional experiments and added some explanations about them. However, the interpretation of some results is not convincing and one of the required experiments has not been done. Therefore, the authors' efforts to improve the manuscript are recognized, but not sufficient.

Specific comments:

[Original comment #2] 

Figure 5: It is desirable to analyze intracellular cAMP levels and Ca2+ efflux, as the authors did in a previous paper. In addition, it is considered that the involvement of cAMP can be clarified by using lysophospholipids in combination with a reagent that lowers cAMP rather than in combination with forskolin that raises cAMP. Similarly, it is desirable to analyze the lysosomal pH.

[Authors’ response #2]

We have previously measured Ca2+ release after treatment with the lysosphingolipids, but the result has been inconclusive. We have also shown in the originally submitted manuscript that the knockdown of P2RX4 did not have any impact on the lysosphingolipid-induced cell death, and we therefore believe that lysosphingolipids does not induce rapid Ca2+ release like CADs. Our data suggest that lysosphingolipids and CADs induce similar manners of cell death, but they don’t induce cell death or lysosomal membrane permeabilization through the exact same mechanisms. Page 8 of 8 We have performed pH measurements, as suggested, to examine whether lysosphingolipids also neutralize lysosomes similarly to CADs prior to lysosomal membrane permeabilization. The lysosphingolipids did not neutralize the lysosome pH after one or five hours of treatments, despite the lysosphingolipids being weak bases like CADs. This new data is presented as a supplementary figure, S1 Fig and the results are described in lines 361-365, page 16: “Unlike CADs (50), the lysosphingolipids did not induce notable elevation in the lysosomal pH in MCF7 cells after one or five hours of treatments and thus prior to lysosomal leakage, when assessed after loading lysosomes with dextran coupled with pH-sensitive fluorescein and pH-insensitive tetramethylrhodamine (S1 Fig).”

[New comments on the authors' response #2]

In panel A, cAMP-dependent CREB phosphorylation was detected in 53 µM GalSph and 40 µM GlcSph, not analyzed in 40 µM GalSph and 28 µM GlcSph. On the other hand, in panel B, the effect of Forskolin on cell death was observed in 40 µM GalSph and 28 µM GlcSph, but not in 43-45 µM GalSph and 30-32 µM GlcSph. From this result, it is presumed that Forskolin does not enhance cell death even at 53 µM GalSph and 40 µM GlcSph. Therefore, from these results, it cannot be determined whether cAMP signaling is involved in the induction of cell death by GalSph or GlcSph. Again, it is thought that the involvement of cAMP can be clarified by using GalSph and GlcSph together with reagents that lower cAMP instead of Forskolin, which raises cAMP. This experiment is easy to try. Additionally, in panel C, the authors also performed P2RX4 knockdown experiments and showed that the induction of cell death by GalSph or GlcSph was not mediated by P2RX4. This experiment showed that the induction of cell death by GalSph or GlcSph was not dependent on the Ca2+ release through P2RX4, but not denied that the Ca2+ release through other channels was involved. Therefore, it is desirable to present the measurement of Ca2+ release as conclusive data. The authors need to explain in more detail, at least not to mislead the reader, even if they do not perform this experiment.

We look forward to receiving your revised manuscript.

Kind regards,

Vladimir Trajkovic

Academic Editor

PLOS ONE

Reviewers' comments:

Reviewer's Responses to Questions

**Comments to the Author**

1. If the authors have adequately addressed your comments raised in a previous round of review and you feel that this manuscript is now acceptable for publication, you may indicate that here to bypass the “Comments to the Author” section, enter your conflict of interest statement in the “Confidential to Editor” section, and submit your "Accept" recommendation.

Reviewer #2: All comments have been addressed

Reviewer #3: (No Response)

2. Is the manuscript technically sound, and do the data support the conclusions?

Reviewer #2: Yes

Reviewer #3: Yes

3. Has the statistical analysis been performed appropriately and rigorously? 

Reviewer #2: Yes

Reviewer #3: Yes

4. Have the authors made all data underlying the findings in their manuscript fully available?

Reviewer #2: Yes

Reviewer #3: Yes

5. Is the manuscript presented in an intelligible fashion and written in standard English?

Reviewer #2: Yes

Reviewer #3: Yes

6. Review Comments to the Author

Reviewer #2: (No Response)

Reviewer #3: In the revised version, the authors have done some additional experiments and added some explanations about them. However, the interpretation of some results is not convincing and one of the required experiments has not been done. Therefore, the authors' efforts to improve the manuscript are recognized, but not sufficient.

Specific comments:

[Original comment #2]

Figure 5: It is desirable to analyze intracellular cAMP levels and Ca2+ efflux, as the authors did in a previous paper. In addition, it is considered that the involvement of cAMP can be clarified by using lysophospholipids in combination with a reagent that lowers cAMP rather than in combination with forskolin that raises cAMP. Similarly, it is desirable to analyze the lysosomal pH.

[Authors’ response #2]

We have previously measured Ca2+ release after treatment with the lysosphingolipids, but the result has been inconclusive. We have also shown in the originally submitted manuscript that the knockdown of P2RX4 did not have any impact on the lysosphingolipid-induced cell death, and we therefore believe that lysosphingolipids does not induce rapid Ca2+ release like CADs. Our data suggest that lysosphingolipids and CADs induce similar manners of cell death, but they don’t induce cell death or lysosomal membrane permeabilization through the exact same mechanisms. Page 8 of 8 We have performed pH measurements, as suggested, to examine whether lysosphingolipids also neutralize lysosomes similarly to CADs prior to lysosomal membrane permeabilization. The lysosphingolipids did not neutralize the lysosome pH after one or five hours of treatments, despite the lysosphingolipids being weak bases like CADs. This new data is presented as a supplementary figure, S1 Fig and the results are described in lines 361-365, page 16: “Unlike CADs (50), the lysosphingolipids did not induce notable elevation in the lysosomal pH in MCF7 cells after one or five hours of treatments and thus prior to lysosomal leakage, when assessed after loading lysosomes with dextran coupled with pH-sensitive fluorescein and pH-insensitive tetramethylrhodamine (S1 Fig).”

[New comments on the authors' response #2]

In panel A, cAMP-dependent CREB phosphorylation was detected in 53 µM GalSph and 40 µM GlcSph, not analyzed in 40 µM GalSph and 28 µM GlcSph. On the other hand, in panel B, the effect of Forskolin on cell death was observed in 40 µM GalSph and 28 µM GlcSph, but not in 43-45 µM GalSph and 30-32 µM GlcSph. From this result, it is presumed that Forskolin does not enhance cell death even at 53 µM GalSph and 40 µM GlcSph. Therefore, from these results, it cannot be determined whether cAMP signaling is involved in the induction of cell death by GalSph or GlcSph. Again, it is thought that the involvement of cAMP can be clarified by using GalSph and GlcSph together with reagents that lower cAMP instead of Forskolin, which raises cAMP. This experiment is easy to try. Additionally, in panel C, the authors also performed P2RX4 knockdown experiments and showed that the induction of cell death by GalSph or GlcSph was not mediated by P2RX4. This experiment showed that the induction of cell death by GalSph or GlcSph was not dependent on the Ca2+ release through P2RX4, but not denied that the Ca2+ release through other channels was involved. Therefore, it is desirable to present the measurement of Ca2+ release as conclusive data. The authors need to explain in more detail, at least not to mislead the reader, even if they do not perform this experiment.

7. PLOS authors have the option to publish the peer review history of their article (what does this mean?). If published, this will include your full peer review and any attached files.

Reviewer #2: No

Reviewer #3: No

---

## [Author Response · Author response to Decision Letter 1]

8 Sep 2022

Comment from Reviewer 3

In the revised version, the authors have done some additional experiments and added some explanations about them. However, the interpretation of some results is not convincing and one of the required experiments has not been done. Therefore, the authors' efforts to improve the manuscript are recognized, but not sufficient.

Specific comments:

[Original comment #2] 

Figure 5: It is desirable to analyze intracellular cAMP levels and Ca2+ efflux, as the authors did in a previous paper. In addition, it is considered that the involvement of cAMP can be clarified by using lysophospholipids in combination with a reagent that lowers cAMP rather than in combination with forskolin that raises cAMP. Similarly, it is desirable to analyze the lysosomal pH.

[Authors’ response #2]

We have previously measured Ca2+ release after treatment with the lysosphingolipids, but the result has been inconclusive. We have also shown in the originally submitted manuscript that the knockdown of P2RX4 did not have any impact on the lysosphingolipid-induced cell death, and we therefore believe that lysosphingolipids does not induce rapid Ca2+ release like CADs. Our data suggest that lysosphingolipids and CADs induce similar manners of cell death, but they don’t induce cell death or lysosomal membrane permeabilization through the exact same mechanisms. Page 8 of 8 We have performed pH measurements, as suggested, to examine whether lysosphingolipids also neutralize lysosomes similarly to CADs prior to lysosomal membrane permeabilization. The lysosphingolipids did not neutralize the lysosome pH after one or five hours of treatments, despite the lysosphingolipids being weak bases like CADs. This new data is presented as a supplementary figure, S1 Fig and the results are described in lines 361-365, page 16: “Unlike CADs (50), the lysosphingolipids did not induce notable elevation in the lysosomal pH in MCF7 cells after one or five hours of treatments and thus prior to lysosomal leakage, when assessed after loading lysosomes with dextran coupled with pH-sensitive fluorescein and pH-insensitive tetramethylrhodamine (S1 Fig).”

[New comments on the authors' response #2] 

In panel A, cAMP-dependent CREB phosphorylation was detected in 53 µM GalSph and 40 µM GlcSph, not analyzed in 40 µM GalSph and 28 µM GlcSph. On the other hand, in panel B, the effect of Forskolin on cell death was observed in 40 µM GalSph and 28 µM GlcSph, but not in 43-45 µM GalSph and 30-32 µM GlcSph. From this result, it is presumed that Forskolin does not enhance cell death even at 53 µM GalSph and 40 µM GlcSph. Therefore, from these results, it cannot be determined whether cAMP signaling is involved in the induction of cell death by GalSph or GlcSph. Again, it is thought that the involvement of cAMP can be clarified by using GalSph and GlcSph together with reagents that lower cAMP instead of Forskolin, which raises cAMP. This experiment is easy to try. Additionally, in panel C, the authors also performed P2RX4 knockdown experiments and showed that the induction of cell death by GalSph or GlcSph was not mediated by P2RX4. This experiment showed that the induction of cell death by GalSph or GlcSph was not dependent on the Ca2+ release through P2RX4, but not denied that the Ca2+ release through other channels was involved. Therefore, it is desirable to present the measurement of Ca2+ release as conclusive data. The authors need to explain in more detail, at least not to mislead the reader, even if they do not perform this experiment.

 

Reply to reviewer’s comment (to ‘New comments on the authors’ response #2’)

We thank the reviewer for constructively commenting on our revised manuscript. We found the comment useful and fully agree. In the previous version of the manuscript, the discussion of presented data regarding the possible involvement of cAMP signalling pathway in the mechanism of cell death induced by lysosphingolipids indeed focused solely on the positive outcome achieved only under one of the tested conditions. We now have extended the discussion on the matter to also include the negative outcome, and most importantly, we write clearly that this matter still remains to be studied. 

We have demonstrated in this manuscript that extracellularly supplied lysosphingolipids can destabilize the lysosomal membranes, similarly to cationic and amphiphilic drugs (CADs). At the same time, the lysosphingolipid-induced cell death is not dependent on Ca2+ release from P2RX4 unlike CADs, and the lysosphingolipids and CADs thus do not induce the same responses of signalling leading to the eventual cell death. We believe that properly revealing the signalling pathways induced by lysosphingolipids would be both challenging and labour-intensive, even though we agree with the reviewer that a few additional simple experiments might answer some of these questions e.g. involvement of cAMP. We therefore find it more suitable to further explore the signalling pathways of lysosphingolipids in other ongoing projects of the lab, and leave the questions not fully answered for the present manuscript. 

We have expanded the discussion section with the following paragraph, in page 24, lines 557-570. Newly added sentences are marked with bold letters; 

“Supporting this idea, we show here that CAD-resistant MCF7 cells are also resistant to the lysosphingolipid-induced cytotoxicity. Additionally, lysosphingolipids activate the cAMP signalling pathway and induce lysosomal leakage as early events occurring prior to cell death, in a manner similar to that observed in CAD-treated cells (22, 28). The role of cAMP in lysosphingolipid-induced cell death is supported by the ability of forskolin to enhance cell death induced by lysosphingolipids at concentrations below their respective IC50 values. On the contrary, forskolin failed to enhance cell death when combined with higher concentrations of lysosphingolipids at around their respective IC50 values, which were used here to demonstrate the phosphorylation of CREB. In addition, the early lysosphingolipid-induced signalling pathway leading to the cAMP accumulation and cell death in MCF7 cells appears different from the P2RX4- and Ca2+-dependent pathway induced by CADs because the depletion of P2RX4 in MCF7 cells had no effect on the lysosphingolipid-induced cytotoxicity. Thus, how lysosphingolipids activate the cAMP signalling pathway and whether it is involved in the cytotoxic mechanism of lysosphingolipids remain to be studied.”

---

## [Decision Letter · Decision Letter 2]

19 Oct 2022

Galactosyl- and glucosylsphingosine induce lysosomal membrane permeabilization and cell death in cancer cells

PONE-D-22-09320R2

Dear Dr. Jäättelä,

We’re pleased to inform you that your manuscript has been judged scientifically suitable for publication and will be formally accepted for publication once it meets all outstanding technical requirements.

Kind regards,

Vladimir Trajkovic

Academic Editor

PLOS ONE

Additional Editor Comments (optional):

Reviewers' comments:

Reviewer's Responses to Questions

**Comments to the Author**

1. If the authors have adequately addressed your comments raised in a previous round of review and you feel that this manuscript is now acceptable for publication, you may indicate that here to bypass the “Comments to the Author” section, enter your conflict of interest statement in the “Confidential to Editor” section, and submit your "Accept" recommendation.

Reviewer #3: (No Response)

2. Is the manuscript technically sound, and do the data support the conclusions?

Reviewer #3: (No Response)

3. Has the statistical analysis been performed appropriately and rigorously? 

Reviewer #3: (No Response)

4. Have the authors made all data underlying the findings in their manuscript fully available?

Reviewer #3: (No Response)

5. Is the manuscript presented in an intelligible fashion and written in standard English?

Reviewer #3: (No Response)

6. Review Comments to the Author

Reviewer #3: (No Response)

7. PLOS authors have the option to publish the peer review history of their article (what does this mean?). If published, this will include your full peer review and any attached files.

Reviewer #3: No

---

## [Editor Report · Acceptance letter]

8 Nov 2022

PONE-D-22-09320R2 

Galactosyl- and glucosylsphingosine induce lysosomal membrane permeabilization and cell death in cancer cells 

Dear Dr. Jäättelä:

I'm pleased to inform you that your manuscript has been deemed suitable for publication in PLOS ONE. Congratulations! Your manuscript is now with our production department. 

Kind regards, 

on behalf of

Prof. Vladimir Trajkovic 

Academic Editor

PLOS ONE